

# Circular Polarization in Atmospheric Aerosols

Santiago Gassó[1,2] , Kirk D.  Knobelspiesse[3]

[1]ESSIC, University of Maryland, College Park, 20740, USA
[2]Code 613, Climate and Radiation Laboratory, GSFC/NASA, Greenbelt, 20771, USA
[3]Code 616, Ocean Ecology Laboratory, GSFC/NASA, Greenbelt, 20771, USA

*Correspondence to*: Santiago Gassó (Santiago.Gasso@nasa.gov)

**Abstract.** Recent technological advances have demonstrated the feasibility of deploying spaceborne optical detectors with full polarimetric capabilities. The measurement of all four Stokes coefficients opens significant new opportunities for atmospheric
aerosol studies and applications. While considerable amounts of attention have been dedicated to sensors with sensitivity to the total intensity and linear polarization (represented by Stokes coefficients I, U, Q), there has been less attention to the additional information brought by measuring circular polarization (coefficient V). This report fills this gap in knowledge by providing an overview of aerosol sources of circular polarization in the atmosphere and discusses possible remote sensing signatures.

In this manuscript, circularly polarized radiation that results from the interaction of incident unpolarized radiation are considered in three physical settings: optical activity originating in biogenic aerosols, alignment of non-spherical particles in the presence of electrical fields (such as dust, smoke, and volcanic ash) and aerosol multiple scattering effects. Observational and theoretical evidence of, and the settings and conditions for non-zero aerosol circular polarization generated from incident unpolarized radiation are here gathered and discussed. In addition, novel radiative transfer simulations are shown to illustrate
notable spectral and other features where circular polarization may provide additional information that is possibly independent from total intensity and linear polarization only observations.

Current techniques for detection of aerosol composition (also referred as aerosol type) from space provide limited information. Remote identification of aerosols such as smoke, volcanic ash, and dust particles can only be accomplished with some degree of confidence for moderate to high concentrations. When the same aerosols are found at lower concentrations (but still high
enough to be of importance for air quality and other concerns), these methods often produce ambiguous results. Thus, the exploration of additional optical techniques is of interest to improve remote detection of aerosol composition. This study is presented as an overview with a goal to provide a new perspective on an overlooked optical property and to trigger interest in further exploration of this subject.

## 1 Introduction

The 1970s were an important decade from the viewpoint of polarimetric remote sensing. The planetary probes Pioneer 10, 11 and 12 surveyed the planets Venus and Jupiter with a full polarimetric sensor (that is, a sensor capable of measuring all





four Stokes parameters) (Murdin, 2001; NASA, n.d.). Since then, no similar probe has been deployed for atmospheric planetary observations. In contrast, technologies based on multispectral (and later multiangle) remote sensing of total intensity have made considerable strides and have been the cornerstone for the current era of global planetary observations of aerosols with

passive sensors. The last twenty years have shown a significant amount of progress in the global quantification of aerosol concentration from space. Passive imagers with observing bands from the near Ultraviolet (UV) to the infrared (IR) range (such as the sensors MODIS/VIIRS, OMI/TropOMI) routinely obtain a global daily distribution of aerosols. The detection methodology is indirect in that measured radiances are analysed via a complex modelling approach to an aerosol optical depth (AOD, representing atmospheric column light extinction) which serves as a proxy of the total aerosol concentration. As a

result, the total amount of aerosols can now be quantified and thus provides an observational constraint for aerosol transport models and aids in decision-making for different fields such air quality and public health (Duncan et al., 2014; Holloway et al., 2021; Mhawish et al., 2018; Sorek-Hamer et al., 2020; Wei et al., 2020).

A notable feature is the discrepancy between the level of compositional aerosol diversity in aerosol models and remote sensing derived aerosol properties.  Aerosol transport models tend to simulate 4-6 aerosol tracers frequently labelled according

to their source type namely smoke, dust and others (Colarco et al., 2010; Inness et al., 2019) whereas transport models with more chemical transformation focus tend to group aerosol types according to composition such as sulphur, soot, etc (Bian et al., 2017; Binkowski and Roselle, 2003). The convergence of these descriptions can be facilitated by observations and thus, the identification of aerosols from space provides an important observational constraint to aid aerosol modelling efforts. However, remote identification of aerosol type (as a proxy for aerosol composition) is still inadequate to address this need.

While existing methods can identify smoke, dust, volcanic ash, and pollution aerosols (Kahn and Gaitley, 2015; Li et al., 2019b; Mhawish et al., 2018), they fail to identify an important aerosol group: biogenic organic aerosols (BOA). These are a subset of a larger group (organic aerosols) and are sourced from precursor gases generated by biogenic activity (e.g. isoprene from forests) or actual biological remains (dead or live bacteria, fungi, spores) injected in the atmosphere (Facchini et al., 2008; Leck and Bigg, 2005; Sanchez et al., 2018; Verdugo et al., 2004). BOAs are an important component in the global

aerosol budget (Myriokefalitakis et al., 2017)  and the contribution to the total aerosol budget is poorly constrained (Alsante et al., 2021). Not only they are effective Cloud Condensation Nuclei (CCN) and Ice Nucleii Particles (Després et al., 2012; O'Dowd et al., 2004; Twohy et al., 2021) but they can be the dominant aerosol in remote marine and tropical environments (McCoy et al., 2015). Notably, there are very few surveys of the abundance of BOA in organic matter aerosols but recent reports highlight their importance in the aerosol budget (Hodzic et al., 2016; Samaké et al., 2019) as well as their presence

throughout the atmospheric column (Perring et al., 2014). They can also be transported long distances and contribute to the intercontinental transport of pathogens (Kellogg and Griffin, 2006).

Current retrieval schemes based on spectral variability of the observed total radiance can confidently distinguish between smoke and dust at moderately to very highly polluted conditions (AOD(500nm) $>\sim$ 0.2- 0.3). However, differentiation of the two types can yield mixed results at moderate concentration levels ($\sim$0.12 < AOD<$\sim$0.23) and detection is highly

unreliable at background conditions (AOD<0.12). These thresholds are approximate because detection effectiveness is highly



dependent on the retrieval assumptions such as surface type and the cloud removal scheme. However, this picture arises from the validation efforts by the different algorithm development teams when trying compare satellite aerosol type proxies as function of aerosol loading with independent observations (Ahn et al., 2014; Franklin et al., 2017; Jethva et al., 2014; Jethva and Torres, 2019; Sayer et al., 2013, 2018; Schutgens et al., 2021). The inadequate detection of aerosol type at moderate and

background concentration of aerosols constitutes an important limitation. For example, aerosols with different composition such biological or sea-salt dominated aerosol in the pristine marine environment (i.e., background conditions) can have very different impacts on cloud formation (Moore et al., 2013) . Thus, having the ability to distinguish between the two would provide important information for cloud model parameterizations and understanding of the evolution of cloud formation in vast areas of, for example, the Southern Ocean. Furthermore, aerosol type discrimination (for example, between pollen, smoke

and dust) at moderate concentration levels is also very important in public health (Magzamen et al., 2021; Manisalidis et al., 2020).

These are features shared by most retrieval schemes using multispectral passive sensor with a single view, such as the algorithms applied to MODIS and VIIRS (Hsu et al., 2013; Levy et al., 2013) . There are, however, more sophisticated sensors that have better aerosol type detection capabilities. For example, angular variability of the spectral radiance (MISR,

Diner et al., 2005), active remote sensing (CALIOP, Winker et al., 2010) and polarization (POLDER,  Herman et al., 2005; Li et al., 2019) are used to better discriminate between aerosol types, but other technical impediments (limited spatial coverage, adequate pixel size) prevent an effective global coverage. Two sensors (POLDER and CALIOP) are highlighted as the first operational sensors to make observations of linearly polarization (LP) of the Earth surface-atmosphere system. The addition of polarization capabilities demonstrated that aerosols can be better identified by their type.  Particle refractive index and

particle shape (proxies for aerosol composition) have been successfully used to determine aerosol types where polarization-insensitive methods have fallen short (Dubovik et al., 2011; Omar et al., 2009; Russell et al., 2014; Stamnes et al., 2018). Like the intensity-only passive sensors, these remote sensing technologies are not very effective for aerosol type identification in low-to-moderate loading conditions.

A notable feature in existing and planned remote sensing sensors using polarization technologies is that they do not

capture all possible optical information from observed upwelling radiances. Specifically, full optical information in an atmospheric signal is represented by the four Stokes components: 1) total radiance (labelled I) is what most passive spectral imagers measure 2) linearly polarized radiances (represented by two components U and Q) such as those used in currently operational (POLDER, CALIOP) and planned (MAIA, HARP2, SPEXone) sensors (Hasekamp et al., 2019; Liu and Diner, 2016; Remer et al., 2019),  3) circularly polarized radiance (V).   While significant attention has been dedicated to studies

involving total intensity and linear polarization, there is a notable dearth of theoretical and observational atmospheric studies focusing on aerosol information contained in the circular polarization (CP) associated with the fourth Stokes element. Much of the seminal work on atmospheric propagation of CP was carried out in the 1970s (Coffeen, 1979; Hansen, 1971; Hansen and Hovenier, 1974; Kawabata et al., 1980; Kawata, 1978) to study the cloudy atmospheres of Jupiter and Venus. However, no additional dedicated work has been carried out afterwards particularly in the context of Earth's atmosphere.





Until recently, the conventional wisdom relied on modelling studies showing that CP is present in multiple scattering environments such as in clouds (Kawata, 1978) or dense smoke clouds (Slonaker et al., 2005). In a multiple scattering setting, the first scattering event generates linearly polarized light (LP) and a particle will scatter circular polarized light in the second scattering event regardless of the particle shape. The magnitude of degree of CP polarization (a common metric for evaluation, defined as DCP=V/I) can be significantly lower than the degree of linear polarization with values in the $10^{-2}$ to $10^{-5}$ range. Its

low range has been a challenging requirement for Earth observing sensors that have exposure times of the order of milliseconds as opposed to planetary probes that had much coarser spatial resolutions and longer exposure times. Sensor sensitivity and lack of a compelling scientific application have been the main reason to justify the lack of inclusion of CP observation in spectro-polarimeters to be deployed in the near future (Craven-Jones et al., 2014; Dubovik et al., 2019; Stamnes et al., 2021; Trippe, 2014). Interestingly this fact was already raised by Kemp et al., (1971) who noted that the "the appreciable linear

polarization of the planets … had discouraged a search for any circular polarization, which is expected to be much smaller than the linear component".

    Innovations in polarimetric hardware and technology now make feasible the deployment of imagers with full-Stokes radiance measurement capabilities (Patty et al., 2019; Rubin et al., 2019; Sparks et al., 2019)  with much easier to deploy and smaller hardware enclosures. This new generation of sensors raises the possibility that in the not-so-distant future there may

be a deployment of an Earth looking sensor with full polarimetric observing technologies. Concurrent with these hardware advances, there has been significant scientific progress in the characterization of polarimetric properties of aerosols. These two developments prompt revisiting the utility of circular polarization from aerosols. Thus, the purpose of this report is to summarize these recent scientific developments and highlight relevant CP features that can be used for better identification of aerosols by remote sensing means, particularly from space. Specifically, three circumstances are identified where circular

polarization is produced in the interaction of sunlight with: 1) Aerosols with chiral molecules (such as pollen and other biogenic aerosols) 2) Aligned non-spherical particles such as those in electric field created within dust, smoke, volcanic ash clouds 3) High concentration aerosols where multiple scattering conditions are present.

    This report is structured as follows. Sections 2 and 3 give introductory information useful for interpretation of the rest of this report. Section 2 provides a brief theoretical introduction on the representation of polarization and some common

assumptions used to simplify computations of polarized scattering. Section 3 introduces the concept of chirality in aerosols and their optical features. Section 4 discusses the alignment of particles in the atmosphere and how it relates to aerosol CP. Section 5 focuses on the evidence available of circular polarization, in laboratory, field and remote sensing studies. Section 6 discusses the considerations needed for studying aerosol propagation in the atmosphere and provides a modelling example of the generation of CP radiances in an ideal atmosphere. Section 7 is a summary of the main features regarding aerosol CP

discussed here with suggestions for future applications and research.



## 2 Theoretical Aspects in the Representation of Circular Polarization

While not attempting to provide a thorough introduction, a few theoretical aspects of CP's representation in the Stokes formalism are highlighted. For an excellent references on the subject, the reader is referred to some of the classic texts (Bohren and Huffman, 1998; Goldstein, 2003; van de Hulst, 1957) from which this section is largely adapted.

A propagating electromagnetic wave can be represented as a wave with its plane of vibration perpendicular to the direction of propagation and this wave vector can be projected onto two perpendicular axes in the same plane. The projected components of the wave in each axis can be slowed down by the solid's inhomogeneities (these inhomogeneities are expressed mathematically as a non-identity dielectric tensor). The difference in propagation speeds in each projection of the travelling wave results in changes in phase and magnitude of each of the observed components of the outgoing light. Elliptical

polarization is the result of this interaction where linear and circular polarization are particular cases. Polarization thus results from the differential speeds in the component of traversing light. The challenge is to relate the observables (i.e. angular and spectrally dependent radiation) with the microphysical structure of the medium.

In the specific the case of incident unpolarized light, CP can arise out of the first scattering event from a single particle if the particle is chiral and oriented and non-spherical. This is the scenario that presents a path for direct aerosol type

identification. In addition, unpolarized light incident upon of an ensemble of particles can result in outgoing CP if multiple scattering occurs in the medium regardless of the particle type and shape. In this case, it is less clear if there is a potential way to link observed CP with the aerosol type under observation or any identifying observation of the system. An expanded description is provided in sections 3, 4 and 6.

The theoretical description of the scattering process can be cast in the Stokes formalism (van de Hulst, 1957). A

polarized beam is represented by a column (or Stokes) vector. The Stokes vectors for the incident and outgoing light are related by the 4x4 scattering matrix ($S$) that represents the transforming medium as:

$$
\begin{bmatrix} I_{out} \\ Q_{out} \\ U_{out} \\ V_{out} \end{bmatrix} = \begin{bmatrix} S_{11} & S_{12} & S_{13} & S_{14} \\ S_{21} & S_{22} & S_{23} & S_{24} \\ S_{31} & S_{32} & S_{33} & S_{34} \\ S_{41} & S_{42} & S_{43} & S_{44} \end{bmatrix} \cdot \begin{bmatrix} I_{in} \\ Q_{in} \\ U_{in} \\ V_{in} \end{bmatrix} \quad (1)
$$

In an experimental setting, the incoming and outgoing light are known information and the above equation is inverted to obtain the elements of matrix $S$. These elements directly depend on particle composition, size, orientation and shape. Composition in optical terms is expressed through the magnitude and spectral dependence of the real and imaginary part of the index of refraction. In turn the latter is related to material elements to the particle through the dielectric matrix (if the solid is inhomogeneous) or constant (homogenous particle).

The elements $S_{4j}$ (bottom row in equation 1) and $S_{i4}$ (right most column) are associated to the circular polarization of an observed particle and can be used as an index to indicate the presence of CP in the observed signal (Bohren and Huffman,





1998; van de Hulst, 1957). For example, with incident unpolarized light (represented as $I_{in}[1,0,0,0]$) in Equation 1, the system will exhibit CP if the matrix has non-zero $S_{4j}$ and $S_{i4}$ that result in an the outgoing vector with non-zero $V_{out}$. From the modelling viewpoint, there are very few analytical solutions of the Maxwell's equations that provide the sixteen $S_{ij}$ coefficients.

For example, exact solutions can be found for $2\pi r/\lambda \ll 1$ and $|mr/\lambda| \ll 1$ (Rayleigh scattering) with r defined as the radius of a particle, m is the particle complex refractive index with respect to the medium, and $\lambda$ is the wavelength of the incident light. For larger particles, analytical solutions exist for homogenous and multilayer spheres, infinite cylinders, certain spheroids and a handful more (Asano, 1979; Asano and Sato, 1980; Bohren and Huffman, 1998). For non-spherical particles, analytical solutions are only computationally effective for small particles due to numerical instabilities for larger particles such as semi-

analytical T-matrix method, and other approximate solutions (Mishchenko et al., 2000). In the case of chiral particles (section 3), there are exact solutions for specific particle shapes.

Another common assumption used to reduce the number of elements in matrix **S** results from the consideration of ensembles of particles. Symmetries in matrix **S** arise when all particles in the ensemble share the same composition, they are randomly oriented, or when particle shapes have mirror symmetry. These three considerations together lead to simplifications

in the scattering matrix by reducing the number of coefficients in matrix **S** (van de Hulst, 1957; Mishchenko and Yurkin, 2017; Perrin, 1942):

$$\begin{bmatrix} S_{11} & S_{12} & 0 & 0 \\ S_{12} & S_{22} & 0 & 0 \\ 0 & 0 & S_{33} & S_{34} \\ 0 & 0 & -S_{34} & S_{44} \end{bmatrix} \quad (2)$$

This matrix has six independent elements, and it is the most common form of matrix **S** used in optical modelling studies of atmospheric aerosols and in the simulation of aerosol radiances for remote sensing applications. Additional simplifications can be achieved by assuming homogenous spheres ($S_{11}=S_{22}$, $S_{33}=S_{44}$) and specific viewing geometries such as backscattering (Mishchenko and Hovenier, 1995). Matrix 2 has been thoroughly used and validated and represents a reasonable assumption when there no a-priori knowledge of the aerosol under observation. For incident unpolarized light, the first scattering event

results in linear and no circular polarized light for particles with scattering properties following Eq. 2. Conversely, when cross-polarization (proportional to $S_{11}-S_{22}$) is detected when measuring angular scattering, the hypothesis of a collection of isotropic spheres can be ruled out (Section 13.6.2 in Bohren & Huffman, 1998).

For groups of particles represented by matrix 2 , non-zero $V_{out}$ can be obtained in the first scattering event only for incident linearly polarized light which generally is not the case for incident sunlight in the Earth's atmosphere. However, in

the near-UV spectral range where Rayleigh scattering is more dominant, this effect could be relevant. In this spectral range, linear polarized light scattered by molecules would provide the incident linear polarization onto the aerosol (this effect is further discussed in Section 6).



Matrix 2 assumes randomly oriented particles, a condition not fulfilled for oriented non-spherical particles. In this case, symmetry of the distribution within the volume leads to 10 independent parameters (Section 5.22 in Van de Hulst (1957) and eq 15.8 in Mishchenko, 2014) , namely:

$$
\begin{bmatrix}
S_{11} & S_{12} & S_{13} & S_{14} \\
S_{12} & S_{22} & S_{23} & S_{24} \\
-S_{13} & -S_{23} & S_{33} & S_{34} \\
S_{14} & S_{24} & -S_{34} & S_{44}
\end{bmatrix} \quad (3)
$$

Similarly, in the case of the presence of ensemble of chiral particles with an excess of right or left particles, 10 coefficients are needed (see Eq.15.10 in Mischenko 2014). If the ensemble is made up of an equal number of such left and right-handed particles, the matrix reduces 6 non-zero coefficients as in matrix 2.

These examples illustrate that for modelling circular polarization of an ensemble of particles, additional scattering matrix elements need to be considered. The next sections will discuss on different elements of evidence showing atmospheric CP and observations of non-zero $S_{i4}$ and $S_{4j}$ elements.

## 3 An Overlooked Source of Circular Polarization: Chiral Aerosols

The term chiral is commonly used within the field of stereochemistry to refer to a molecule that cannot be superimposed to its own mirror image. Such molecules are called isomers or chiral (hand in Greek). The distinction is important because while the molecule and its mirror image have the same atoms and bonds, they may exhibit different chemical reactions and distinctive optical properties because the orientation of their atoms. Non-biogenic chiral molecules are present in equal quantities of the two orientations (referred as left or right isomers, figure 1), but biogenic chiral molecules are only present in one orientation, that is, there is no natural occurring molecule of the opposite orientation. The orientation in space is important because it determines how the molecule interacts with incident electromagnetic radiation. Therefore, aerosols that contain biological material also contain chiral molecules and have singular optical properties that result from the interaction with light. When unpolarized or linearly polarized light shines on a chiral particle, it will scatter circularly polarized light (known as optical rotation) or absorb polarized light (known as circular dichroism) such that the outgoing light is circularly polarized. Such molecules are said to exhibit optical activity. A particle such as dust will not generate circularly polarized light in the first interaction with light because it is not optically active. If there are equal number of left and right handed molecules (referred as racemic mixtures), the two resulting opposite circular polarizations will cancel each other. The mixtures with excess of one isomer can occur naturally or in synthetic compounds such as pharmaceuticals. For example, in the 1960s the use of thalidomide as a morning sickness drug by pregnant women resulted in severe birth defects. It was later found that thalidomide was a racemic mixture where the R-enantiomer had the intended sedative relief, whereas the S-enantiomer had a



negative impact on fetal development (Kim and Scialli, 2011; Smith, 2009). Another example of chiral molecule is the common analgesic Ibuprofen (Shi et al., 2021) : only the S form of the molecule is used for therapeutic applications (Evans, 2001; Neupert et al., 1997). In the natural world, chiral molecules are present in commonly found BOA such as α-pinene. A recent study (Bellcross et al., 2021) showed that different tacticity (that is, arrangement of chiral centres) of α-pinene dimers may influence its physical properties, such as hygroscopicity thus impacting cloud droplet formation (Cash et al., 2016).

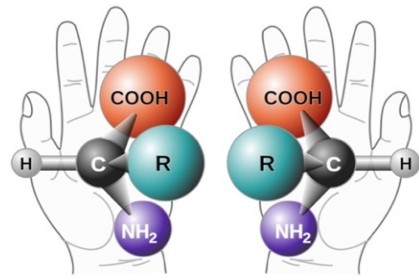

**Figure 1**: Two isomers (or enanatiomers) of a generic amino acid, both molecules have same composition but different molecules orientation, which determines the polarization of the scattered light. Biological proteins are made of exclusively on left-handed amino acids and right-handed sugars. (Source: Wikipedia Commons)

There is a rich history around the discovery and development of the science of chiral molecules. Detailed summaries can be found in Barron (2009), Lakhtakia, (1990) and Applequist, (1987). A few historical developments in the history of optical activity are highlighted here. For example, in 1812, optical activity was first reported by Jean Batiste Biot and in 1848, Louis Pasteur established that chiral compounds have distinctive chemical properties, and the optical activity was associated with the molecule's properties. A model of the physical interaction between the incident light and a chiral molecule was first proposed in 1915 by Max Born and Carl Oseen and in parallel by Frank Gray (Applequist, 1987) . By representing the incident wave as a combination of left and right circular polarized waves (Born, 1918; Drude et al., 1902), the solutions of the scattering matrix were found by requiring two different indices of refraction to be associated to the chiral particle (one index for each right and left circular polarization component). The different indices of refraction arise after imposing the condition of non-diagonal dielectric constant resulting in a nonlinear constitutive equation (eq 8.5 in Bohren & Huffman, 1998). Exact solutions for chiral particles were found for the case of homogenous (Bohren, 1974) and layered spheres (Bohren, 1975).

Applequist (1977) provided a graphic explanation on how this physical interaction operates. When incident polarized light is modelled as a combination of two (right and left) circular polarized waves, each wave interacts with each atom by inducing a dipole (black arrows in Figure 2). The strength and direction of each dipole is directly dependent on the rotation of the incident wave and the orientation of the molecule polarizability at that point. The total induced dipole that is, the aggregated effect of all molecules' induced dipoles, will be different for each of the incident waves. A different total dipole for each wave results in a polarizability tensor that is not diagonal, and results in a non-linear change in the constitutive relationship between the displacement and electric field vectors (see equation 8.5 in Bohren and Huffman, 1998). The direct consequence of this effect is that two different indices of refraction need to be considered for each wave to obtain a solution in the Maxwell equations. The difference between the two results in the optical rotation effect.





Optical activity is observed in biological and non-biological compounds. However, non-biological OA materials have

equal number of left and right molecules and generally have no net CP. Notably, biological chiral molecules are present in one type of handedness and in non-racemic mixtures which result in distinctive CP. Chiral molecules are present in biogenic aerosols and provide a path for their detection via optical methods as it is discussed next.

In addition to pharmacology, optical studies of chiral molecules have been carried out in several fields for different applications. In molecular biology, optical polarimetry is a common technique because it is possible to non-destructively

identify tissues, molecules, and cells. There have been significant developments in the use of optical activity to achieve this (Ghosh, 2011; Li et al., 2018; Purvinis et al., 2011; Qi and Elson, 2017; Westphal et al., 2016). Approaches were developed to measure and model the polarization signatures of biological particles in turbid media (Nafie, 1995; Sloot et al., 1989). Also, numerous works on instrumentation (Ahn et al., 2011; Bickel et al., 1976; Keller et al., 1985; Kunnen et al., 2015), theoretical and computational approaches (Autschbach, 2009; Ghosh, 2011; Ghosh et al., 2008; Videen, 1998; Wang et al., 2002; Wood

et al., 2007) and reviews (de Boer and Milner, 2002; Savenkov, 2011; Tinoco and Williams, 1984) for applications in the medical sciences have been carried out. It is notable the amount of effort centred on the study of biological particles and tissues by means of measuring the full scattering matrix elements.

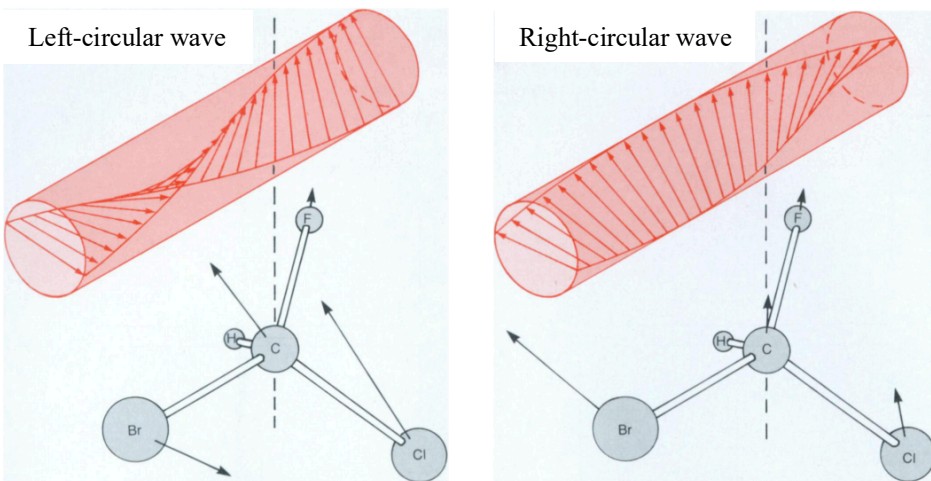

Figure 2: Illustration of the interaction of incident light decomposed in two orthogonal components (represented by the left and right circular waves) with the chiral molecule CHFClBr. As the wave moves through the molecule, it induces a dipole in each atom (arrows). The aggregate total induced dipole in the molecule is different for each incident wave, which in turn results in different refractive indices for each wave. This difference produces optical activity which can be observed in the laboratory (Figure from Applequist , 1987)

CP detection is also established in astronomy (Bailey et al., 1998; Meierhenrich et al., 2002; Rosenbush et al., 2007). The concept of chiral molecules have been invoked as a possible tracer of biomolecules in comets and extra-solar planets

(Patty et al., 2018; MacDermott, 1997; Gledhill et al., 2007; Degtjarev & Kolokolova, 1992). Excess of L(or Left)- amino acids have been found in meteorites (Cronin and Pizzarello, 1997; Glavin and Dworkin, 2009). They showed that indeed homochirality exists outside the Earth in materials known to be fundamental for biological systems. The search for chiral





signatures in meteorites remains an active area of research (Avnir, 2021; Glavin et al., 2020b, 2020a) and is a candidate technique to detect biosignatures in future probes to Encedalus (MacKenzie et al., 2021; Neveu et al., 2020). Remote sensing
of CP has been proposed as technique for detection in comets and planets (Kolokolova et al., 2011). Measurements of CP from comets using spectropolarimeters have also been reported and chiral particles were suggested as a possible source of the observed CP (Rosenbush et al., 2007). Remote sensing of CP in space requires highly sensitive measurements (fractional polarizations well below $10^{-4}$). However, modern developments in design and technology breakthroughs are making the detection of such signals possible (Craven-Jones et al., 2014; van Harten et al., 2011; Hough, 2011; Hough et al., 2006; Patty
et al., 2017; Tyo et al., 2006).

These developments show that circular polarization is used to understand chiral particles in several diverse fields and is becoming increasingly feasible with technological advancement.

## 4 Alignment of Particle as a Source of Circular Polarization

As noted in section 2, matrix 2 is a commonly used form of the scattering matrix with some null off diagonal elements.
The reasons invoked to set the coefficients $S_{41}=S_{42}=S_{14}=S_{24}=0$ are the assumptions of randomly oriented particles and presence of equal number of mirror particles in the volume under observation. This results in a lower number of matrix coefficients to compute thus achieving much faster results. Thus, modelled scattering by such aerosols with incident unpolarized light would result in no circular polarization in the first scattering event according to this formulation. It is the thesis of this work that some of these assumptions need to be reassessed considering additional evidence regarding aerosol morphology and their
occurrence in the environment.

Two of the main aerosol types (dust and smoke) in the atmosphere are known to be non-spherical at different stages while airborne. For example, freshly formed smoke contains abundant chains of coagulated soot particles (Chakrabarty et al., 2014; China et al., 2013; Girotto et al., 2018). A smoke cloud eventually ages, and a significant portion of particles become more spherical as they react with sunlight and water and organic vapours condense. Dust particles are also non-spherical (Gao
and Anderson, 2001; Muñoz et al., 2001; Okada et al., 2001) and while different modelling approaches exist for the computation of their optical properties, they are expensive to run and only recently have satellite retrieval algorithms been incorporating non-spherical dust models (Gassó and Torres, 2016; Zhou et al., 2020). While the computation of matrix elements for single particles of these aerosol types will result in non-zero $S_{34}$ and $S_{43}$, their impact in the ensemble of particles cancel out because the assumption of randomly oriented particles, which obscures the individual effects of particle shape
(Mishchenko, 2014). However, in the case of oriented non-spherical particles such as dust aligned with an electromagnetic field (figure 3), the random particle orientation assumption is not fulfilled and the elements $S_{41}$, $S_{42}$, $S_{14}$, $S_{24}$ should be non-zero. In this case, the particle shape favours charge separation, and the resulting induced dipole in each particle aligns with the electrical field. This concept has been applied in astronomy studies where the observations of linear and circular polarization in comets and interstellar dust could not be properly reproduced by models with spherical particles. Rather, non-spherical dust


models were required to successfully reproduce the observations (Gledhill and McCall, 2000; Rogers and Martin, 1979; Schmidt, 1973; Vandenbroucke et al., 2021; Whitney and Wolff, 2002). Further, an interplanetary dust study by Kolokolova & Nagdimunov (2014) calculated the LP and CP in ensembles of spheroids oriented in a magnetic field and reported non-zero values for both. Notably they reported magnitudes of degree of CP comparable to degree of LP for alignment angles between the major axis and the existing field between 0 and 90 degrees.

With regards to Earth sciences applications, there is growing body of evidence indicating that the presence of electrical fields in volcanic ash (Harrison et al., 2010; Lane et al., 2011) and dust layers (Kamra, 1972; Nicoll et al., 2010; Ulanowski et al., 2007). Also, intense electrical fields are present in large fires so strong that they generate their own lighting (Dowdy et al., 2017; LaRoche and Lang, 2017). In these cases, it is conceivable to consider the presence of alignment of smoke particles immediately downwind from a fire.

There are additional interesting features to note. While it's been known that electrical fields are present during dust

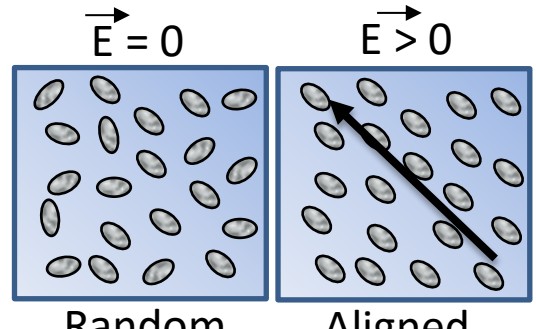

Figure 3: Dust particles in suspension are normally assumed as randomly distributed (right panel). But in the presence of an electric field, they tend to orient along (or precess around) the direction of the field

emission and play a role in the amount of dust lifted (Esposito et al., 2016; Kok and Renno, 2008; Zhang and Zhou, 2020) , particles remain charged during long distance transport and persist for significant distances. This includes reports of electrical fields in Saharan dust clouds over the UK (Harrison et al., 2018) and in Greece (Daskalopoulou et al., 2021). The presence of an electric field may influence the natural sedimentation process of the particle and may provide an explanation to unexplained

observations of very large particles found in dust clouds thousands of kilometres from the source (Denjean et al., 2016; Maring, 2003; Ryder et al., 2012) . As suggested by Ulanowski et al (2007), the electrical field can impact the aerodynamics of large particles by counteracting gravitational sedimentation and cause a preferential orientation in non-spherical particles (Mallios et al., 2021; Toth III et al., 2020).

    In summary, circular polarization has been observed originating in interstellar dust. Only by assuming particle

alignment in the presence of an electromagnetic field the optical properties can be modelled to match observations. Since similar mechanisms of particle orientation are present in the Earth's atmosphere, and it may well be possible that these oriented particles scatter circular polarized light as well.

## 5 Observations of Aerosol Circular Polarization

Most circular polarization studies in the Earth's environment have been limited to applications using active sensors

or in the laboratory. Lidars with CP detection capabilities have been proposed for cloud phase and particle shape detection (Donovan et al., 2015; Nicolet et al., 2012; Cao et al., 2009; Chaikovskaya, 2008; Hu et al., 2003). Similarly, CP detection using lidars has been proposed to remotely detect bio-warfare agents (Pendleton and Rosen, 1998; Rosen, 1993) and to improve visibility and detection in underwater turbid environments (Gilbert and Pernicka, 1967; Lewis et al., 1999) and foggy atmospheres (van der Laan et al., 2017). These studies highlighted the fact that in high- and low-density particle environments,

the propagation of circular polarization does not degrade as quickly as linear polarization.

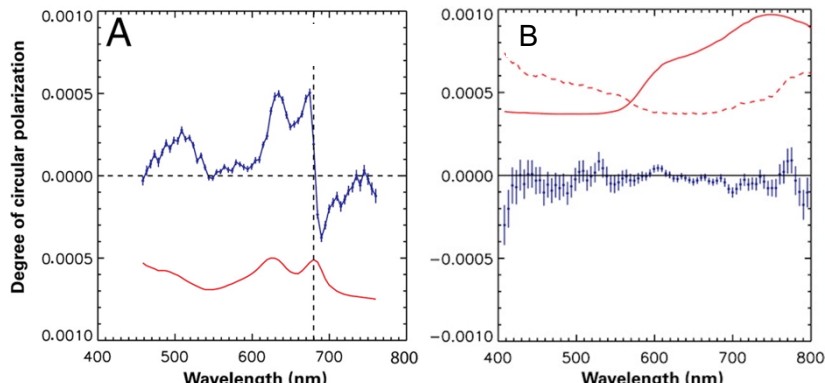

Figure 4: Non-zero and near zero transmission CP (solid blue line) spectra of a cyanobacteria (A) and from a control mineral (B). The solid red line in A is a scaled version of the absorbance spectrum. Vertical dashed line points the reversal in sign in CP. Solid red in B shows the reflection spectrum of iron oxide, and the dashed red line shows the degree of linear polarization, both arbitrarily scaled (Sparks et al, 2009).

Recent developments show that CP can be measured with high precision using spectro-polarimetric techniques (Sparks et al, 2009). Martin et al. (2010) measured the CP scattered by biogenic and dust aerosols and illustrated the difference in the matrix element $S_{14}$ in those aerosols. The magnitude of particle CP (expressed as the ratio $S_{14}/S_{11}$) can vary significantly with values as high as 0.02 reported for particles in suspension or even higher for individual particles (Pan et al., 2022; Shapiro

et al., 1990) or they can be as low as $10^{-5}$ (Sparks et al., 2011). Figures 4 (Sparks et al., 2009) show measurements of CP transmission spectra in phytoplankton and a non-biogenic aerosol (dust). Figure 4b confirms earlier observations of lack of CP in dust and volcanic rocks reported first in Pospergelis (1969). The Sparks et al. (2009) study demonstrated the clear contrast in CP when compared with non-biological materials. As shown, the spectral CP is wavelength dependent, with a dramatic sign reversal at roughly 680nm (marked with a vertical dashed line). This is a phenomenon referred as the Cotton effect and it is a

known biogenic spectral signature (Barron, 2009).

The above-mentioned studies (Martin et al., 2016; Sparks et al., 2009) and others (Patty et al., 2018; Nouri et al., 2018) have used spectropolarimeters in laboratory settings with enough sensitivity ($<10^{-4}$) to detect DCP signals from biogenic aerosols. These technologies not only have demonstrated that the full Stokes vector can be remotely measured with passive





sensors but are amenable to be adapted in more portable forms (Rubin et al., 2019) or are already being built for aircraft
deployments (e.g. Patty et al., 2021).

There are very few reports of measurements of circular polarization specifically addressing atmospheric aerosols.
Perhaps one of the most representative were made by Bickel et al (1976) who reports with a technique to measure all
polarization states of light scattered by atmospheric aerosols. They showed several examples of non-zero circular polarization
in pollen and bacterial spores as a function of scattering angle. Figure 5 shows an example of circular polarization scattered

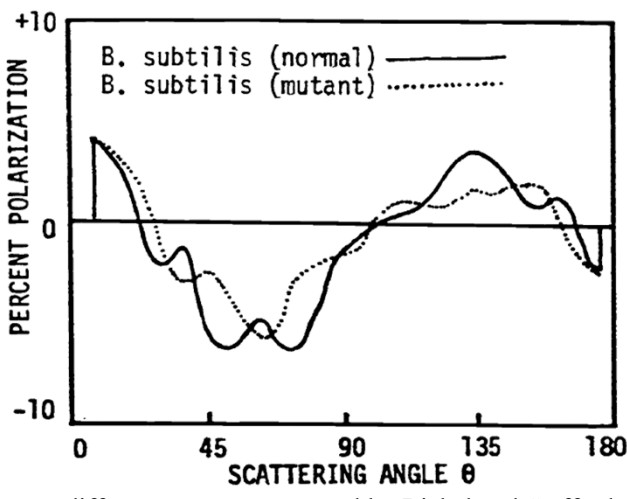

**Figure 5.** Spores are known biogenic aerosols and commonly found in the atmosphere and as this example illustrates, they scatter circular polarized light. This figure shows the scattering matrix element measured one type of spores and its mutant form for incident linearly polarized light. The variability of $S^*_{34}$ (ratio of $(S_{14}+S_{34})$ /$(S_{11}+S_{31})$ ) as a function of angle is displayed. (Bickel and Stafford ,1996)

by two different spores as reported by Bickel and Stafford (1996) . Because these observations, Bohren & Huffman, (1998)
suggested (section 13.8.5) the use of $S_{34}$ measurements as a way to detect biological particles. Further this is not only the only
non-zero element in the scattering matrix. Laboratory measurements of CP of biological particles confirm that the element $S_{14}$
is non-zero for individual (Lofftus et al., 1988; Pan et al., 2022) and ensembles of particles (Shapiro et al., 1990), although in
the latter the magnitude tends to be lower than for single particles. This is an intriguing aspect that should be further explored
as a non-zero $S_{14}$ element implies that circular polarization should be present with incident unpolarized light such as sunlight.
This can be easily realized by multiplying the incident sunlight vector ($I_{sun}$=[I,0,0,0]) by equation 1.

Additional studies reported optical properties of hydrosols (a.k.a aerosol in aqueous environments) such as
chloroplasts (Gregory and Raps, 1974) and dinoflagellates (Liu & Kattawar, 2013; Shapiro et al., 1990, 1991). They measured
CP in commonly found marine aerosols (a.k.a hydrosols). Hydrosols are precursors of atmospheric biogenic organic aerosols
(BOA) in the marine environment, and they are not unique in exhibiting chiral signatures. For example, organic films
containing amino acids attach to ejected marine aerosols and were detected and classified according to their chiral structure
(Kuznetsova et al., 2005; Wedyan and Preston, 2008). Further, they do have spectro-polarimetric signatures.





As far as specific reports of observations of CP in atmospheric aerosols or presence of aerosols with chiral elements, the available evidence does indicate that biogenic aerosols contain chiral materials which suggests they should have distinctive

CP signatures. Although polarized light has been proposed as a tool to improve visibility observation in the atmospheric marine boundary layer (Quinby-Hunt et al., 1997), the concept of CP detection was seemingly overlooked. In contrast, several investigations on the chiral nature of commonly found atmospheric aerosols have become available in the last decade. For example, naturally formed secondary organic aerosols (SOA) have a distinct chiral structure in the Amazon forests (Ebben et al., 2011; Martinez et al., 2011) and in maritime pine forests (Staudt et al., 2019) . The fact that aerosol gaseous precursors are

chiral is important because the chirality of the gas remains when it forms or attaches to an aerosol as shown by Ebben et al.,(2012). Enantiomeric monoterpenes (another biogenic gas emitted by plants, (Ganjitabar et al., 2018) and a precursor of SOA) have been measured in forests of South America and northern Europe (Song et al., 2014; Williams et al., 2007; Yassaa et al., 2001). Isoprene , another known aerosol precursor, is also produced in the marine environment by phytoplankton and is associated with the formation of BOAs in the atmosphere above (Colomb et al., 2008). It has been found in different enantiomer

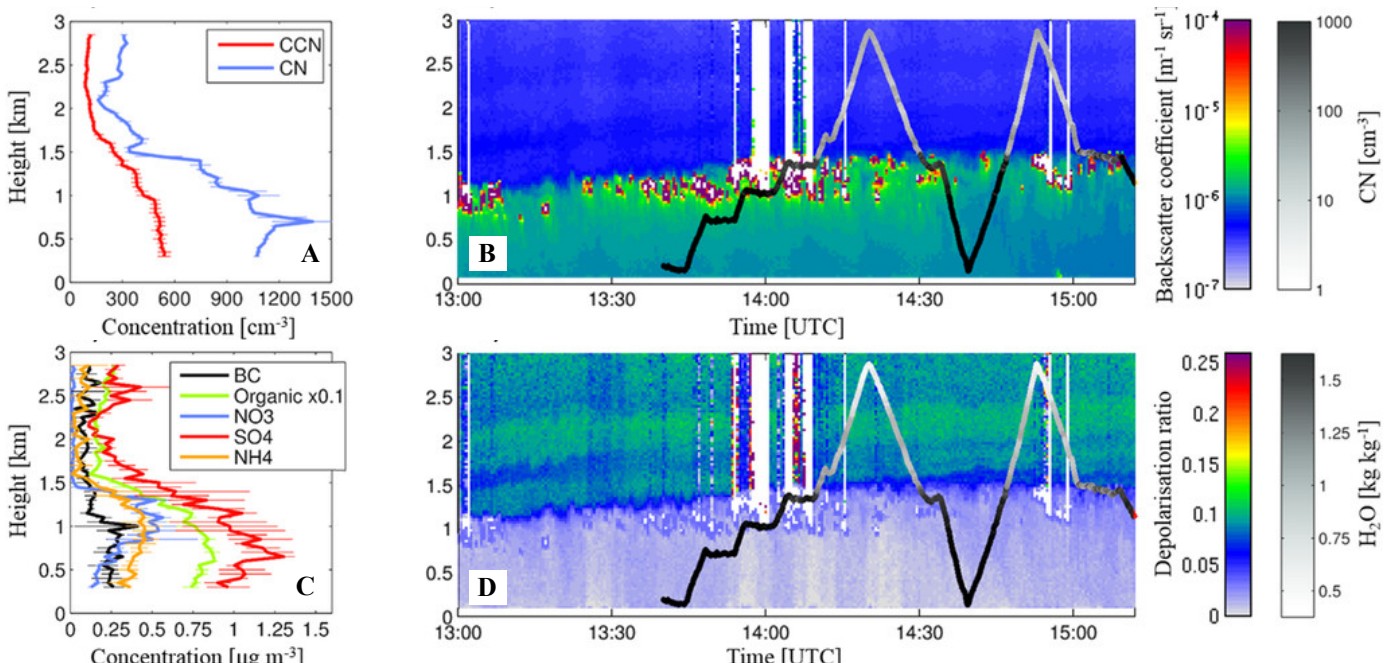

**Figure 6:** During Springtime, forests are known to contain significant quantitates of biogenic organic aerosols (such as pollen) and are good candidates for detection of circular polarization. Panel D shows non-zero CP above a forest in Finland where particle concentrations are rather high (measured by a nearby aircraft , A and B) and aerosol composition consistent with organic aerosols (C). Panels B and D include airplane aerosol concentration and ambient water vapor (from Petäjä et al., 2016)

forms in ambient aerosols and used as a tracer to determine whether an aerosol is of primary or secondary origin (Noziere et al., 2011; Cash et al., 2016). Also, the chiral nature of a biogenic aerosol decays with aging through oxidation (Salma et al., 2010) suggesting that chirality detection may be used as a marker for aerosol aging.



The presence of left and right amino acids has been used for tracing the aerosol origin of marine aerosols and to determine their age in the Antarctic atmosphere (Barbaro et al., 2015; Kuznetsova et al., 2005). Also, circular dichroism has

been observed in aerosol water extracts and atmospheric humic substances (HULIS) obtained from PM2.5 um samples in rural, urban and in tropical environments, and has been suggested as a possible way to differentiate the aerosol between anthropogenic and natural formation processes (Salma et al., 2010). Gonzalez et al., (2014) determined the distinctive chirality in BOAs to distinguish between aerosols of primary and secondary origin. Chiral behaviour in aerosols has not only been reported over land but also in the marine environment (Yassaa et al., 2008). Further, glucose is one of the main components

found in all biogenic organic aerosols (Samaké et al., 2019) and given that glucose is a chiral molecule and naturally present as one type of isomer (D-(+)-glucose), focusing efforts to detect CP from this molecule in BOA seems appropriate.

Perhaps one of the more compelling observations of aerosol's circular polarization in the Earth's atmosphere is reported by Petäjä et al., (2016) . The BAECC (Biogenic Aerosols—Effects on Clouds and Climate) Campaign was carried out during Spring and Summer, 2014 in the Finish arctic forest, and it focused on characterizing the role of biogenic aerosols

in cloud formation. The campaign included several examples of collocated surface lidar observations and aircraft overpasses at a time of the year where pollen and other biogenic aerosols are abundant. Figure 6 shows an example from this study where distinctive CP (here defined as ratio of left circular return divided by right circular return) was measured by ground-based HSRL polarization lidar with in-situ confirmation of high aerosol concentrations and composition consistent with organic aerosols in the boundary layer (Lee et al., 2018).

Interestingly the distinctive CP found by lidar is in agreement with the theoretical modelling study by Kolokolova & Nagdimunov, (2014) where optically active particles were shown to have non-zero CP and zero linear polarization in the backscattering direction. However, a controlled study (Cao et al., 2011) measuring the degree of LP and CP in pollen backscattering found that both scale with each other following the predictions of Mishchenko & Hovenier, (1995). That study concluded there is no additional aerosol information by measuring both LP and CP. While these two offer somewhat

conflicting conclusions, both highlight that indeed biogenic aerosols do produce circular polarization. Clearly additional studies need to be carried out on the independence of information brought by the fourth Stokes term.

Overall, these measurements show that chirality detection is useful not only for aerosol type identification but also for determination of particle creation mechanism and particle type. However, while these studies stress the fact that atmospheric aerosols do contain OA components, they are aerosol in-situ composition and identification studies and to the

extent of this search, there are no studies focusing on the spectral or polarimetric properties and the optical features of bulk chiral aerosols in the atmosphere.

## 6 Circular Polarization Propagation in the Atmosphere

The study of circular polarization propagation has received significant attention outside the Earth sciences. For example, there is considerable amount of work in the field of astronomy to identify interstellar dust in the presence of magnetic


fields (Section 3). In addition, there has been significant theoretical and modelling work towards understanding the propagation of CP in dense media such as tissues or dense biological liquids (Autschbach, 2009; Ghosh, 2011; Ghosh et al., 2008; Videen, 1998; Wang et al., 2002; Wood et al., 2007). This contrasts with the handful of theoretical works published on radiative transfer that considered oriented non-spherical particles or chiral particles in low density medium such as an Earth-like atmosphere. Probably the only work on the development of the radiative transfer equations for chiral media in a planetary

atmosphere were the studies by Kokhanovsky (1999, 2002). Specially, Kokhanovsky (1999) found the solution of the vector radiative transfer equation in the single scattering approximation, but not additional work has been published with this line of research. The design of a radiative transfer code that ingests chiral or non-symmetric oriented particles scattering matrix requires the treatment of a scattering matrix with two angles (scattering and inclination) as well as consideration of extinction matrix. To date, there are no algorithmic approaches available that consider these phenomena (Ping Yang, personal

communication). However, such tools do exist for spherical homogenous particles, and it is with one of these that the top-of-the-atmosphere main features of aerosol CP are illustrated. The next section presents a modelling study with idealized conditions to simulate propagation of CP in an Earth-like atmosphere.

**6.1 An application: CP from multiple scattering in an ideal atmosphere**

Multiple scattering by an ensemble of particles can result in circular polarization (Kawata, 1978; Slonaker et al., 2005). Equation 4 shows the resulting 4th Stokes vector element (V) after two scattering events for incident unpolarized light onto an ensemble of randomly oriented particles with equal number of mirror symmetric particles (see Appendix 1 for derivation):

$$V_{out} = I_{in}.s_{12}.S_{34}\big[\cos\big(2(\pi - m_2)\big).\sin\big(2(-M_1)\big) + \sin\big(2(\pi - m_2)\big).\cos\big(2(-M_1)\big)\big] \quad (4)$$


where $V_{out}$ and $I_{in}$ are components of the outgoing and incident Stokes's vector, $s_{12}$ and $m_2$ are scattering matrix elements and meridian angle (defined in figure A1) corresponding to the first scattering event, $S_{34}$ and $M_1$ are the terms for the second scattering event. In this case, the first interaction results in linear polarization which becomes the incident radiation for the second scattering event.

As an illustration on how the different components of the Stokes vector vary as a function of aerosol size distribution, composition, and concentration, radiative transfer computations of the top-of-the-atmosphere (TOA) radiances at a fixed geometry for selected aerosol models are shown. The goal is to illustrate how the degree of linear and circular polarization compares with each other in ideal viewing conditions such as in the case of aerosols over a dark surface.

The simulated case assumes the following conditions: aerosols with very low to high concentrations, homogenously

vertical distributed from the surface to 1km, and bi-lognormal size distributions with varying coarse mode concentrations. The details are as follows. The size distribution is made of two lognormal distributions with a variable fine mode fraction (0<FMF<1) defined as in dV/dlogR = FMF * dVfine/dlogR  + (1-FMF) dVcoarse/dlogR with  $R_{vol,fine}$ = 0.1165 um, $STD_{fine}$





=1.4813 , $R_{vol,coarse}$= 2.8329 um, $STD_{coarse}$=1.9078 (based on Dubovik et al., 2002, JAS). With respect to the refractive index, three composition models are considered. One model (labeled "Low") has n= $1.50-i1.0x10^{-4}$ and the other model ("High")

$1.50-i1.0x10^{-3}$ for all wavelengths in both models. A third model ("Mixed") has a constant real part with spectrally dependent imaginary refractive index and different composition in each mode: Fine mode= $1.45+i 5.0x10^{-3}$ to $2.0x10^{-4}$ and coarse mode = $1.50 + i1.0x10^{-4}$ to $1.0x10^{-6}$. The spectral range chosen includes an assortment of wavelengths from the near UV to near-IR (in um): 0.3400, 0.3880, 0.4700, 0.7650, 2.210. The particle shape is assumed to be spherical. The atmospheric column only contains air (i.e., no trace gases) with a surface pressure set at 1013mb, surface reflectance is set to zero, sun zenith angle is

40 degrees with variable azimuth and view angles set at regular spaces. Aerosol loadings range from AOD(0.551 um) from 0 to 1.2. The only radiative processes considered are aerosol and Rayleigh scattering. The solar constant is set to unity. The radiative transfer model is a Gauss-Seidel vector code ("the Arizona Code") created by Herman et al., (1995) and is frequently used in near-UV satellite applications (Herman et al., 2001; Herman & Celarier, 1997).The code outputs the four Stokes components of the upwelling radiance (I, Q, U and V) and then the degree of Linear polarization (DLP  = $SQRT(Q^2 + U^2)/I$

) and circular polarization (DCP= $|V|)/I$) are computed.

## 6.2 Results

Figure 7 provides an example of these simulations. Three panel show the total intensity, DLP and DCP as a function of aerosol loading (AOD at 0.551 µm) at 5 representative wavelengths for an aerosol model dominated by fine mode particles

(FMF=0.98). Before pointing out features to note, it is useful to reiterate a few concepts here. The Rayleigh molecular scattering cross section model is strongly dependent on the inverse fourth power of the wavelength, and it has a minimal influence on scattering in high wavelengths (roughly λ > 0.6 µm). Thus, the interplay of aerosol and molecular scattering will be strongly dependent on the region of the spectrum. Also, multiple scattering is strongly dependent on particle concentration and its impact on observed parameters will be more apparent at higher AODs.

For no aerosols (AOD=0), no circular polarization is present since molecular scattering only results in linear polarization. The intensity (left panel) remains constant at low AODs (AOD<0.1) and then increasing with aerosol loading mostly in the visible and near-UV channels because most of the scattering is occurring in the fine mode which is the dominant aerosol concentration in this simulation (the impact of the presence of a variable coarse mode is explored later).

In considering DLP (center panel), the effect of multiple scattering is a strong function of aerosol loading and

wavelength. Below 0.5 µm, molecular scattering provides a background of linearly polarized light incident on the aerosol. This effect is most apparent at these wavelengths even in the low AOD range. As aerosol concentration increases, the number of scattering events between Rayleigh and aerosols also increase. Linear polarization is an expression of the prevalence of single scattering aerosol events which increases with loading. However, at a certain loading level, the number of second scattering events start to be dominant and linear polarization degrades due to cancellation. This is particularly apparent in the lowest



three wavelengths where multiple scattering is dominant at all aerosol loadings. At the higher wavelengths (0.76 and 2.2 µm), the Rayleigh contribution is minimal, and the only linear polarization observed is originating from primary aerosol scattering. Aerosol-to-aerosol scattering interactions are prevalent at these wavelengths and aerosol concentration becomes the dominant effect. In the case of the 0.76 µm wavelength, a second scattering event starts to be dominant at AOD ~ 0.4 and the contribution of LP to the observed signal is dampened.  At 2.2 µm, single scattering dominates through the AOD range considered.

480          The degree of circular polarization (DCP) is shown in the right panel of figure 7. There are several features to note. First, with no aerosols (AOD=0), there is no circular polarization present as expected. Second, while the absolute magnitudes of DCP are low compared to DLP, the change with loading is significant. Third, there is a strong wavelength dependency in the DCP magnitude. There is no DCP at the 2.21 µm wavelength and it is highest at the VIS and NIR wavelengths (0.47 and 0.76 µm) and intermediate for the near-UV wavelengths. This suggests an interplay between the presence of Rayleigh effect

in the different spectral ranges. At 2.21 µm, scattering is only originating from aerosols and possibly there are not enough scattering events to produce meaningful LP and subsequent second scattering events to generate CP. At the lowest wavelength (0.34 µm, blue line), LP from Rayleigh scattering is the dominant initial source of polarization, and the addition of aerosols

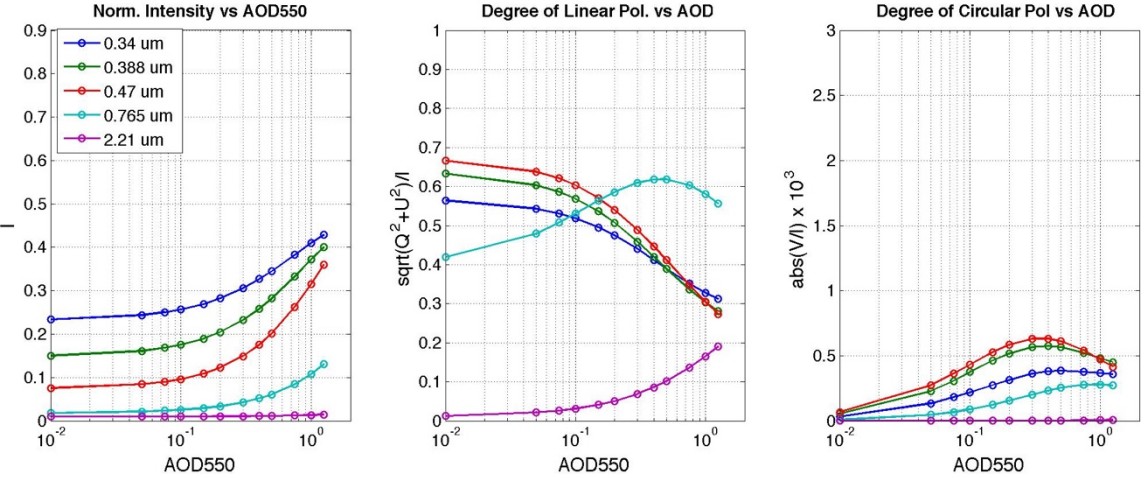

Figure 7: Top-of-the-Atmosphere normalized radiance (left), degree of linear (center) and circular (right, multiplied by $10^3$) polarization as a function of aerosol optical depth (AOD) at 550um for selected near-UV to near-IR wavelengths at scattering angle 160.8 degrees (SZA=40, VZA=40, RelAzim=30). This is the mixed aerosol model with high concentration of particles in the fine mode and a minimal contribution of the coarse mode (FMF=0.98).

has an immediate multiple scattering effect in diminishing the DLP (center panel) and resulting in more second scattering events.  The resulting increase in CP with loading appears to be limited (it flattens around AOD=0.5). The variability in LP

and CP for the intermediate wavelengths (0.380, 0.47 and 0.765 µm) is more difficult to interpret but it is clear it results from the interplay of Rayleigh scattering dependence on wavelength and aerosol concentration available enabling different orders of scattering.





The magnitude of both DLP and DCP as a function of loading are strongly dependent on the viewing geometry whereas it is less so for total intensity. For example, figure 8 shows Intensity, DLP and DCP for the same aerosol model and

conditions of figure 7 but for a different scattering angle. In comparison with figure 7, the magnitude of the intensity increases slightly, and it does not vary significantly with a change of geometry. However, DLP is reduced by a factor of 3-4 whereas DCP is augmented by a factor 2-3. Also, the shape of the DLP curves appear dampened in figure 8 whereas the slope of the DCP curves is markedly steeper.  Finally, at both geometries the same range of wavelengths (0.38 and 0.47 μm) had the highest CP magnitudes. The contrast in polarization features between figures 7 and 8 illustrates the importance of viewing geometry

in polarimetric observations.

The role of the dominance of coarse and fine mode particles and composition are evaluated next. These simulations were carried out with a goal to explore if compositional differences (expressed optically through different refractive indexes and size distribution) arise in the Stokes vector components. This analysis focuses on low aerosol loading ranges ($\tau_{aer}$ = 0 to 0.2) because this is the range of concentrations most globally prevalent and for which current approaches have the least

sensitivity to aerosol type. In all cases, the pure Rayleigh atmosphere ($\tau_{aer}$ =0.0) is included as a reference point.

These simulations are shown in figure 9 at two representative wavelengths. The intensity plots (panels in the first column) show an expected behaviour. In the near UV, molecular scattering is dominant and the changes in aerosol loading, and fine mode contribution add minor increases compared to the baseline. Also, no differences are noted due to aerosol

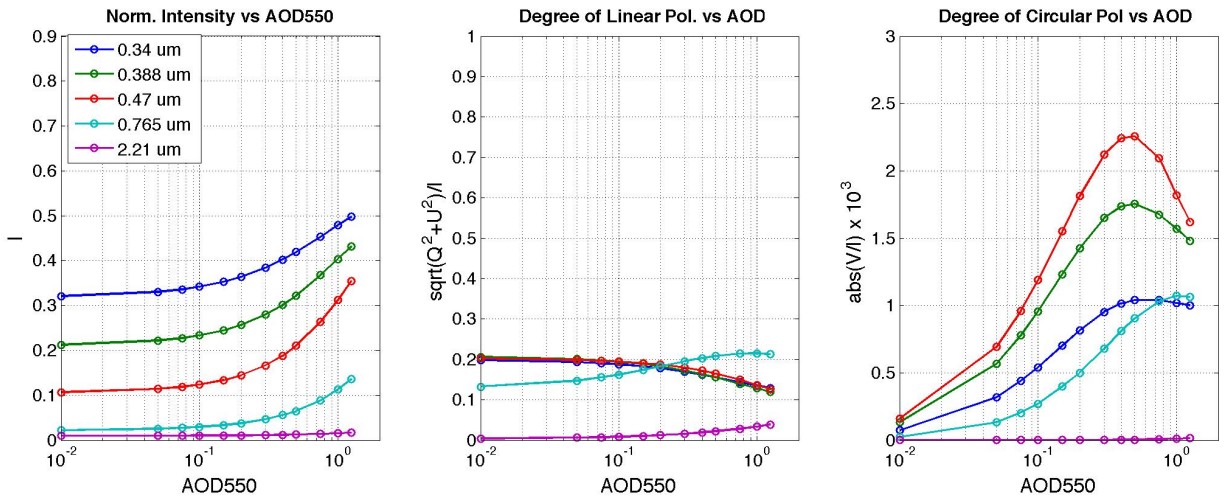

Figure 8: Top-of-the-Atmosphere normalized radiance (left), degree of linear (center) and circular (right, multiplied by $10^3$) polarization as a function of aerosol optical depth (AOD) at 550um for same aerosol properties as figure 7 but with a different scattering angle 112 degrees (SZA=40, VZA=40, RelAzim=120)

composition. At 0.865 μm, the Rayleigh contribution can be ignored and, given that the surface is dark, the signal originates

only from aerosols. Note the significant difference in magnitude as well as the clear increase of radiance as a function of





aerosol concentration as expected in the single scattering regime. Both panels illustrate the lack of aerosol identifying information in intensity only observations at these aerosol loading ranges.

The linear (center column panels) and circular (right) polarization features are discussed next. DLP has a more noticeable dependency on the contribution of FMF. For the near UV, multiple scattering is dominant resulting in immediate

decrease of linear polarization as function of FMF and loading because both tend to increase the number of scattering events. This is not the case at 0.865 μm where linear polarization increases with both AOD and FMF. Note the contrast in magnitude changes in both wavelengths. In practical terms, the relative changes in DLP magnitude at 0.380μm is not significant as function of loading, composition and FMF. But at 0.865μm, there is notable change with AOD and FMF but negligible for composition. For circular polarization, it is notable the large change in absolute and relative magnitude at both wavelengths as

function of FMF and it is more pronounced at higher loadings. However, the sensitivity to composition is not particularly noticeable at these loadings.

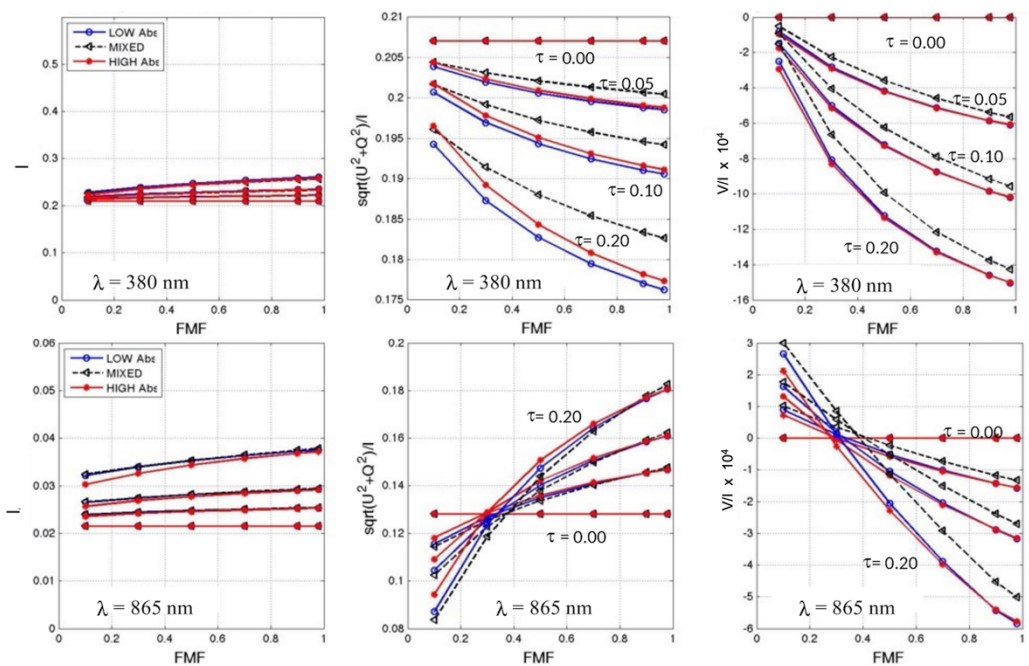

Figure 9: Intensity (left column), degree of linear polarization (center) and normalized circular polarization (right) as a function of Fine Mode Fraction (FMF) for 3 aerosol models with different imaginary refractive index: model with Low and High spectrally invariant absorption and a case with spectrally dependent absorption (Mixed). See section 6.1 for details. Angles are the same as in figure 8.

Overall, these plots suggest there is a sensitivity to differences between coarse and fine mode dominated aerosols in both linear and circular polarization. In terms of absolute and relative magnitude change and considering both wavelengths,





circular polarization is more sensitive to modal variability. More importantly both linear and circular polarization demonstrate
a sensitivity to aerosol properties that are not present in the intensity only plots.

While not comprehensive, these simulations illustrate the patterns in circular polarization for representative aerosol loadings, composition, and size distributions. They demonstrate that CP can be highly variable and has variable magnitude as a function of aerosol properties in conditions where LP does not change much (figures 8 and 9). It demonstrates that CP may contain aerosol information in instances when LP loses its sensitivity to aerosol properties as aerosol optical depth increases.
A more formal way to determine the unique value of CP contained in observed polarized radiances would require an information content assessment that incorporates all observations simultaneously and accounts for expectations of model and measurement uncertainty. Successful studies for this application have used the Bayesian locally linear approximation method of Rodgers (Hasekamp and Landgraf, 2007), or a full Bayesian approach (Knobelspiesse et al., 2021).

## 7 Summary and Recommendations

It has been more than forty years since polarimetric sensors were deployed to study planetary atmospheres: Pioneer 10 and 11 sent to Jupiter in 1972 and 1973,respectively and Pioneer 12 was sent to Venus in 1978 (Murdin, 2001; NASA, n.d.). All three missions carried out the same imaging polarimeter (Gehrels et al., 1980). Their observations resulted in several novel computations and theoretical studies focused on full polarization in cloudy atmospheres. These studies suggested that while circular polarization (CP) originating from clouds was observed, no significant additional information appear to be
gained by measuring the fourth Stokes coefficient. These conclusions, combined with the low magnitude of observed circular polarization compared to linear polarization and total intensity, resulted in little subsequent interest in the development of full polarization sensors for remote sensing applications.

Five decades later, technological advances now enable a wider use of circular polarization in several applications including remote detection of aerosols. Additionally, there is new knowledge of sources of CP in the atmosphere. It is now
reasonable then to re-assess the advantages of carrying out full polarization remote sensing of atmospheric aerosols. The purpose of this overview is to provide a summary of these recent developments. This report is concerned with the science motivations regarding aerosol properties and points out features that are likely (or already known) to produce circular polarization.

Three aerosol physical processes are explored as a source of circular polarization in the atmosphere:
1) Optical activity in biogenic aerosols (Section 3 and 5)

2) Alignment of non-spherical particles such as dust, volcanic ash, and smoke (Section 4)

3) Multiple scattering effects of aerosols (Section 6)

Observable CP in biogenic particles originates from molecules such as amino acids and proteins, both known to be chiral molecules with specific optical features and suggest a pathway to uniquely differentiate biogenic aerosols from non-
biological aerosols. The presence of atmospheric electrical fields within clouds of non-spherical aerosols results in an





alignment of particles and may provide asymmetries in the scattering process that may result in CP. While the presence of strong electrical fields in dust, volcanic ash and smoke has been observed, circularly scattered light resulting from incidence of (unpolarized) sunlight remains to be observed. Finally, CP can arise because of multiple scattering processes in any group of particles.

The latter effect is further explored by carrying radiative transfer simulations for a typical set of aerosol models at different viewing geometries, wavelength, and aerosol loadings in idealized conditions (dark surface, no trace gases, and no clouds, only in a molecular atmosphere). The total intensity, the degrees of linear and circular polarization are compared against each other and features and differences among the three are highlighted. CP variability is noted in scenarios where total intensity and linear polarization do not appear to be as sensitive as a function of angle, loading and spectral range. CP does
degrade as a function of aerosol concentration, but not as much as linear polarization does. This may provide a pathway by which CP and LP can be used in combination to extract different aerosol information depending on the aerosol loading. As is common in polarization, CP is highly sensitive to viewing geometry and additional studies remain to be done to explore this aspect in relation to satellite observing geometries.

         The above simulations are meant to provide limited illustration and extrapolation to more realistic settings need to be
considered. Clearly there are important challenges to be sorted out in a realistic atmospheric setting. Observation of circular polarization requires precise and accurate measurements where instrument sensitivity and additional sources of instrumental noise need to be properly characterized. Moreover, there are additional (non-aerosol) sources of CP in the atmosphere that need to be considered and compared to those arising from the particles of interest. For example, circular polarization is contained in the incident solar radiation at the Earth's Top-of-the-Atmosphere (DCP $\sim 10^{-6}$) (Kemp et al., 1987). Also circular
polarization is expected to arise from the surface such as vegetation (Van Eeckhout et al., 2019; Patty et al., 2019) . However, there have been no attempts to simulate scenarios with all these elements in place in a radiative transfer model and establish the different contributions to the top-of-atmosphere polarized radiances. Perhaps, the closest to these scenarios are several studies in object detection in foggy or very turbid (including underwater) environments in the context of military applications. In general, these studies are interested in horizontal transmissivity of spectro-polarimetric signals and they do highlight the
persistence of CP compared to LP (van der Laan et al., 2018; Zeng et al., 2018; Zhang et al., 2019, 2020) . However, none of these studies considered incident of unpolarized light or background conditions such as those in the atmospheric remote sensing setting.

         This is the first work to summarize circular polarization in atmospheric aerosols. It gathers evidence from different disciplines (molecular biology, stereochemistry, astronomy, and astrobiology) where aerosol CP is present, and it discusses
the possibilities of carrying out similar observations in atmospheric aerosols. Specifically, it offers a possibility by which atmospheric aerosols can be detected remotely and may provide a new tool to complement existing techniques of aerosol composition identification.



## Acknowledgements

The work represents the culmination of an idea that originated in 2009 during NASA's working groups to develop the scientific objectives of the ACE mission (later evolved into current PACE and ATMOS mission). Thanks to program managers Hal Maring and Paula Botempi for their initial funding. The stimulating conversations through the intervening years with Bill Sparks, Robert Spurr, Craig Bohren and Ludmilla Kolokova are also acknowledged as seminal to this work. Bill Sparks and Noah Rubin are thanked for the comments of early versions of the manuscript. Finally, SG is deeply thankful to Kerry Meyer

(NASA/GSFC) who provided the funding for the compilation and completion of the work shown herein.

## Data availability

The data from the model simulations and the implemented model codes are available from the authors upon request.


## Authors contribution

SG carried literature gathering and review, performed the simulations, carried out the analysis, created the figures and wrote the paper. KDK reviewed several versions of the manuscript, provided several key suggestions and background material.


## Competing interests

The authors have declared no competing interests.





**Appendix A**

Derivation of the 4ᵗʰ Stokes vector component in the case of two scattering events for an aerosol particle with the following scattering matrix

$$S(\Theta) = \begin{bmatrix} S_{11} & S_{12} & 0 & 0 \\ S_{12} & S_{22} & 0 & 0 \\ 0 & 0 & S_{33} & S_{34} \\ 0 & 0 & -S_{34} & S_{44} \end{bmatrix}$$

where the dependence on the scattering angle $\Theta$ (a function of the incident and viewing zenith ($\theta,\theta'$) and azimuth angles ($\mu,\mu'$) ) in $S_{ij}$ is assumed and omitted for simplicity.

The transformation of coordinates (Chandresekar, 1950) to the local meridian planes is (as defined in figure A1)

$$L(\Theta) = R(\pi-m_2) . S(\Theta) . R(-m_1) \qquad (A1)$$

where R is the rotation matrix transforming from the particle centred frame of reference to the observer's frame of reference
and is defined as:

$$R(\psi) = \begin{bmatrix} 1 & 0 & 0 & 0 \\ 0 & \cos(2\psi) & \sin(2\psi) & 0 \\ 0 & -\sin(2\psi) & \cos(2\psi) & 0 \\ 0 & 0 & 0 & 1 \end{bmatrix}$$

with $\psi = \pi - m_2$ or $= -m_1$, $m_1$ and $m_2$ are the angles between the meridian planes containing incident and scattered vectors and the scattering plane (figure A1), respectively.

For incident unpolarized light ($I_{in}[1,0,0,0]$) , the outgoing scattered radiation after the first scattering event can be expressed as (after equation 1 in Section 2):

$$\begin{bmatrix} I_{in}S_{11} \\ I_{in}S_{12}\cos(2(\pi-m_2)) \\ -I_{in}S_{12}\sin(2(\pi-m_2)) \\ 0 \end{bmatrix} \qquad (A2)$$

The resulting vector after the first interaction has linear components that depends on the orientation of the reference planes, and it has no circular polarization ($V_{out}=0$). If there is a second scattering event, the above vector (A2) is the incident light onto the second particle and a similar multiplication of matrixes (Eq A1) occurs resulting in:

$$V_{out} = I_{in} . s_{12} . S_{34}\left[\cos(2(\pi-m_2)) . \sin(2(-M_1) + \sin(2(\pi-m_2)) . \cos(2(-M_1))\right]$$


Where the upper-case letters refer to the second scattering event and the lower case to the first event.





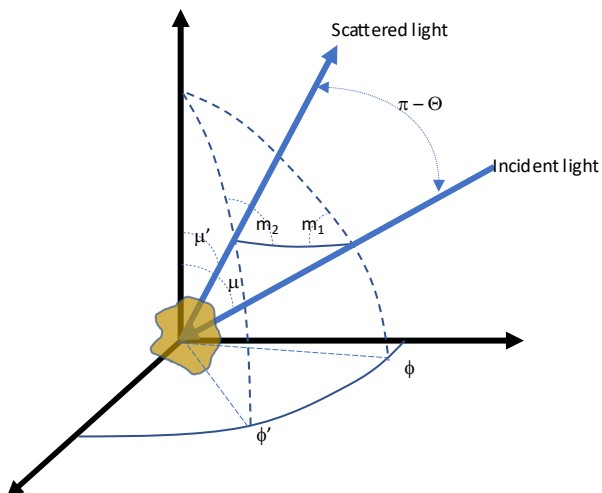

Figure A1: Angle definitions in a frame of reference centred in the scattering object: $\mu,\phi$ ($\mu'$,$\phi'$) are the zenith and azimuth for the incident (scattered) light, $\Theta$ is the scattering angle defined by the plane that contains the incident and scattered light vectors, $m_1$ and $m_2$ are the angles between meridian planes and the scattering plane.



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
