# Peer review of "Circular Polarization in Atmospheric Aerosols"

_Atmospheric Chemistry and Physics, 2022_

## Author Comment (AC1)

Reply to R1

We thank R1 for the editing suggestions and insightful comments. Some of them prompted us to go back and think on aspects of the manuscript that we did not think about. So, we appreciate the intellectually stimulating commentary. We addressed all the points and provided answers or just commentary to some of the points that invited discussion. We did modify the manuscript based on the feedback and we believe the manuscript has been strengthened . We hope the R1 agrees.

**Lines 22-28, "Current techniques… this subject.": Include also the reasoning for using circular polarization measurements for better typing/characterization at low concentrations.**

We made a small change in the abstract to address this concern as we did not want to lengthen the abstract too much.

OLD:
When the same aerosols are found at lower concentrations (but still high enough to be of importance for air quality and other concerns), these methods often produce ambiguous results. Thus, the exploration of additional optical techniques is of interest to improve remote detection of aerosol composition. This study is presented as an overview with a goal to provide a new perspective on an overlooked optical property and to trigger interest in further exploration of this subject.

NEW

When the same aerosols are found at lower concentrations (but still high enough to be of importance for air quality and **cloud formation**), these methods often produce ambiguous results. The circular polarization of aerosols is rarely utilized, and we explore its value for improved determination aerosol composition. This study is presented as an overview with a goal to provide a new perspective on an overlooked optical property and to trigger interest in further exploration of this subject.

**Lines 39-42, "As a… Wei et al., 2020)": Please rephrase.**

We rephrased the sentence. Now it reads

OLD:
As a result, the total amount of aerosols can now be quantified and thus provides an observational constraint for aerosol transport models and aids in decision-making for different fields such air quality and public health (Duncan et al., 2014; Holloway et al., 2021; Mhawish et al., 2018; Sorek-Hamer et al., 2020; Wei et al., 2020)

NEW:

"With this retrieved quantity, it is now possible to quantify with satellite observations the total amount of atmospheric aerosols globally. Aerosol global transport models now have such observations available and are used to constraint model initializations or validate model outputs. In addition, they contribute to more focused and applied decision-making fields such air quality and public health (Duncan et al., 2014; Holloway et al., 2021; Mhawish et al., 2018; Sorek-Hamer et al., 2020; Wei et al., 2020)"

**Lines 50-51, "While existing… (BOA)": Line 49 states that the "remote identification is still inadequate…". Then you state that the "existing methods can identify smoke etc..". Please rephrase, or provide a measure of uncertainty for the results of the "existing methods".**

We choose to rephrase the sentence and now it reads as less controversial.

OLD:
"While existing methods can identify smoke, dust, volcanic ash, and pollution aerosols (Kahn and Gaitley, 2015; Li et al., 2019b; Mhawish et al., 2018), they fail to identify an important aerosol group: biogenic organic aerosols (BOA)."

NEW:
"Standard aerosol detection techniques can identify several aerosol types (here interpreted as a proxy for aerosol composition) such as smoke, dust, volcanic ash, and pollution aerosols (Kahn and Gaitley, 2015; Li et al., 2019b; Mhawish et al., 2018). However, an important aerosol group is not detected by these techniques: biogenic organic aerosols (BOA)."

**Line 71, "...such biological...": Replace with "...such as biological...".**

Done

**Line 75, "…important in public health…": Replace with "…important for public health…".**

Done

**Lines 73-74, "These are… Levy et al., 2013)": Move this phrase to the previous paragraph.**

Done

**Line 82, "adequate pixel size": Do you mean "non-adequate pixel size"?**

Replaced with "a large pixel size"

**Lines 86-88, "Like the intensity-only… loading conditions": Please provide reference(s) for this statement.**

We recognize the sentence is a bit too opinionated in that it is more based on personal experience and partial evidence. A brief review of available information does appear to support the statement. Here are a few facts:

Aeronet retrievals of refractive index only occur at AOD(440)>0.4 because this is the minimum threshold advised by the Aeronet team for reliable composition retrievals. When no microphysical properties are available (known as Inversion data in Aeronet terminology), a dedicated study using Aeronet direct retrievals (Sayer et al 2012) recommended to use a screen selection of AOD<0.2 and AE<1.0 for marine aerosol detection. Russell et al (2014) introduced a methodology for aerosol typing identification. They presented an aerosol typing scheme using linear polarization from the satellite sensor POLDER 3 over the ocean and successfully identified aerosol types at lower than 0.2 AOD magnitude. However, the methodology was applied to one region over the ocean and this approach does not appear to be of regular use in the community. For MISR, the recommended threshold of aerosol type identification is 0.2 (Kahn and Gaitley, 2015). Another good example is provided by the Schutgen et al (2021) study where several satellite groups contributed to the analysis. This analysis included data from both intensity only and intensity + linear polarization observations. The goal was to evaluate and compare satellite observations of aerosol type proxies. To have certainty that the satellite obs provided a "truth" against which models could be compared, the authors chose to exclude satellite retrievals with AOD<0.3 . The same authors explore the choice of threshold to find the right value which depended on the sensor, retrievals algorithm used , surface and number of comparison points (See figure 12 in Schutgen et al (2021). Future sensors with LP measurement are expected to perform better at distinguishing aerosols but the uncertainties associate to intense properties to identify aerosol type vary. For example, the Knobelspiesse et al, (2012) modeling studying showed that the retrieval uncertainty in imaginary index refraction is significant for AOD <~0.2 although in the case of single scattering albedo (SSA) appears to be low enough for distinguishing among aerosol groups.

Overall, because there is no single aerosol composition marker in measured radiances, current aerosol operational algorithms identify aerosol groups indirectly. Such labeling arises from spectral and angular variability  (and in some cases linear polarization magnitude) observed and an aerosol type is selected following a decision tree or statistical method in the respective algorithms. Generally, aerosol type (or aerosol model selected by the algorithm) is reported in the output file. In those cases where the algorithm developer team describe the product, they recommend using a threshold in AOD for reliable aerosol group identification.

We find that properly backing up this statement may imply to add too much information to support it and since it is not essential, we slightly modified the paragraph to make it less controversial.

Russell, P. B., Kacenelenbogen, M., Livingston, J. M., Hasekamp, O. P., Burton, S. P., Schuster, G. L., et al. (2014). A multiparameter aerosol classification method and its application to retrievals from spaceborne polarimetry. Journal of Geophysical Research: Atmospheres, 119(16), 9838–9863. https://doi.org/10.1002/2013JD021411

Schutgens, N., Dubovik, O., Hasekamp, O., Torres, O., Jethva, H., Leonard, P. J. T., Litvinov, P., Redemann, J., Shinozuka, Y., de Leeuw, G., Kinne, S., Popp, T., Schulz, M., and Stier, P.: AEROCOM and AEROSAT AAOD and SSA study – Part 1: Evaluation and intercomparison of satellite measurements, Atmos. Chem. Phys., 21, 6895–6917, https://doi.org/10.5194/acp-21-6895-2021, 2021.

Sayer, A. M., Smirnov, A., Hsu, N. C., & Holben, B. N. (2012). A pure marine aerosol model, for use in remote sensing applications. Journal of Geophysical Research: Atmospheres, 117(D5), n/a-n/a. https://doi.org/https://doi.org/10.1029/2011JD016689

OLD :

[revised manuscript text omitted]

Done

**Lines 168-171, "For non-spherical particles,… particle shapes.": Include discussion on the numerical solutions as well (e.g. ADDA calculations for dust particles in Gasteiger et al. (2011)).**

Done, Added the Gateiger et al and Kempinnen et al references.

**Lines 189-192, "However,… Section 6).": Rephrase, referring to this light as multiple-scattered light.**

Done

**Lines 199-201, "Similarly… in matrix 2.": Provide the scattering matrices for the ensemble of chiral particles, and the ensemble of chiral particles that is made up of an equal number of left- and right-handed particles.**

Done. We added the matrix for the case of ensemble of chiral particles with an excess of left or right particles. For equal number is the same as Matrix 2.

**Lines 238-239, "Exact solutions… (Bohren, 1975).": This is confusing… Can chiral particles have spherical shapes? Please clarify here.**

Yes, chiral particles can be spherical as shown by Bohren. The chirality arises because asymmetry in the molecules that make the solid sphere. This asymmetry is expressed in the dielectric constant for the molecule. For non-chiral particles, the dielectric constant is assumed the same for all directions in the solid (ie $\varepsilon_{xx}=\varepsilon_{yy}=\varepsilon_{zz}=\varepsilon_{ij}$). The connection with the bulk optical properties of the solid is through the dielectric constant which is directly related to the square of module of the refractive index. In matrix representation, the dielectric constant is represented by an identity matrix multiplied by $\varepsilon_o$. In the case of a chiral particle, $\varepsilon_{xx}$ is not equal to $\varepsilon_{yy}$ and the optical property of the solid is represented by a non-identity matrix with at least two different coefficients. Bohren (1975) shows that this directional asymmetry in the solid results in two different indexes of refraction associated to this asymmetry in the dielectric constant representation. In practical terms, when using Bohren scattering code for chiral particles, the user has to input two refractive indexes.

We added the following few words in the respective paragraph that we hope makes it a bit clearer and avoids a lengthy explanation:

OLD:
" Exact solutions for chiral particles were found for the case of homogenous (Bohren, 1974) and layered spheres (Bohren, 1975)."

NEW

"Exact solutions **for solid particles made of chiral molecules** were found for the case of homogenous (Bohren, 1974) and layered spheres (Bohren, 1975)."

**Line 263, "Rosenbush et al., 2007": This work is not included in the list of references. Please add.**

Added

**Line 286, "Two of the main … while airborne.": Include also the marine particles (e.g. Haarig et al., 2019) and pollen.**

Done

OLD
Two of the main aerosol types (dust and smoke) in the atmosphere are known to be non-spherical at different stages while airborne. For example, freshly formed smoke contains

abundant chains of coagulated soot particles (Chakrabarty et al., 2014; China et al., 2013; Girotto et al., 2018).
NEW
There are aerosol types known to be non-spherical at different stages while airborne. For example, pollen shape inhomogeneity and dry marine aerosols result in detectable linear polarization (Haarig et al,2019). Also freshly formed smoke contains abundant chains of coagulated soot particles (Chakrabarty et al.,2014; China et al., 2013; Girotto et al., 2018).

**Line 297-298, "In this case,… electrical field.": Provide reference.**

Done. Added Mallios et al, 2021 reference

Mallios, S. A., Daskalopoulou, V., & Amiridis, V. (2021). Orientation of non spherical prolate dust particles moving vertically in the Earth's atmosphere. Journal of Aerosol Science, 151, 105657. https://doi.org/https://doi.org/10.1016/j.jaerosci.2020.105657

**Lines 301-304, "Further… 90 degrees.": Kolokolova and Nagdimunov (2014) also proposed a methodology for differentiating aligned particles from optically-active particles. Please include this info here, as well.**

The methodology proposed by KN2014 to differentiate aligned particles from chiral particles is mentioned in a later section 5 where actual observations of chiral particle are discussed.

**Lines 308-309, "In these cases… a fire.": Provide reference, or explain more, to support for this statement.**

This comment prompted us to search for more evidence on the presence of electrical fields in in smoke from fires. We found a few references (all from Russian origin, we did not find similar studies carried out in the Western hemisphere) reporting ambient electrical field variability associated with smoke from forest fires. So, we rewrote the paragraph as follows:

OLD:
" Also, intense electrical fields are present in large fires so strong that they generate their own lighting (Dowdy et al., 2017; LaRoche and Lang, 2017). In these cases, it is conceivable to consider the presence of alignment of smoke particles immediately downwind from a fire."

NEW:
"Further, smoke from fires have associated variability in the local atmospheric electrical field as it has been observed in Siberia (Ippolitov et al., 2013; Phalagov et al., 2009). This variability was associated to changes in aerosol load, ambient moisture and actinic flux (Nagorskiy et al., 2014). The presence of electrical fields and the fact that fresh smoke particles are non-spherical aggregates suggests the possibility of charge distribution within particles and possible alignment with the ambient electrical field "

**Lines 312-313, "This includes... (Daskalopoulou et al., 2021).": The studies of Harrison et al. (2018) and Daskalopoulou et al. (2021) do not prove that the particles passively "remain charged" far from the sources. Instead, Harrison et al. (2018) refers to "an active charging process" and provides the triboelectrification as a possible cause. Please include more info here from the respective papers.**

We modified the paragraph as suggested:

OLD
",particles remain charged during long distance transport and persist for significant distances. This includes reports of electrical fields in Saharan dust clouds over the UK (Harrison et al., 2018) and in Greece (Daskalopoulou et al., 2021)."

NEW
"There are additional interesting features to note. While it's been known that electrical fields are present during dust emission and play a role in the amount of dust lifted (Esposito et al., 2016; Kok and Renno, 2008; Zhang and Zhou, 2020), non-background electrical fields are also found in dust clouds at significant distance from the source, such as dust reaching Greece and the UK (Harrison et al., 2018, Daskalopoulou et al., 2021 ). These studies suggested triboelectrification (i.e. friction between particles) as mechanism for generation of an electric field within the cloud during transit."

**Lines 320-322, "Since... as well.": Replace with "Since similar mechanisms of particle orientation seem to be present in the Earth's atmosphere, and it may well be possible that these oriented particles scatter circular polarized light as well."**

DONE

**Lines 320-321, "Since... atmosphere.": Support this statement with more info.**

Since both this comment and the previous one is similar, we rephrased the sentence hoping that it addresses both points:

OLD
"In summary, circular polarization has been observed originating in interstellar dust. Only by assuming particle alignment in the presence of an electromagnetic field the optical properties can be modelled to match observations. Since similar mechanisms of particle orientation are present in the Earth's atmosphere, and it may well be possible that these oriented particles scatter circular polarized light as well."
NEW
"In summary, circular polarization has been observed originating in interstellar dust and in this case only by assuming particle alignment in the presence of an electromagnetic field the optical

properties can be modelled to match observations. Since electrical fields are present in dust clouds in the Earth's atmosphere, it may well be possible that these oriented particles scatter circular polarized light as well."

**Lines 325-326, "Lidars… Hu et al., 2003).": Include also the polarization lidars discussed in Paschou et al. (2022) and Tsekeri et al. (2021).**

DONE

**Lines 331-333, "Recent… aerosols.": There is no "Martin et al. (2010)" in the reference list. If you mean the work of Martin et al. (2016), they do not provide measurements of mineral aerosol. Please correct.**

DONE

**Line 350, " …by two different spores as reported by…": Replace with " …by two different spores illuminated with linearly-polarized light, as reported by…".**

DONE

**Lines 351-352, "Further… matrix": Replace with "Further this is not the only non-zero element $S_{4j}$ or $S_{i4}$ of the scattering matrix.".**

DONE

**Line 357, "Additional studies…": Do you refer to laboratory studies? Please clarify.**

We modified the paragraph slightly to address this point:

OLD:
"Additional studies reported optical properties of hydrosols (a.k.a aerosol in aqueous environments) such as chloroplasts (Gregory and Raps, 1974) and dinoflagellates (Liu & Kattawar, 2013; Shapiro et al., 1990, 1991). They measured CP in commonly found marine aerosols (a.k.a hydrosols). Hydrosols are precursors of atmospheric biogenic organic aerosols (BOA) in the marine environment, and they are not unique in exhibiting chiral signatures. For example,…"

NEW:
"Additional laboratory studies reported optical properties, including CP, of hydrosols (a.k.a aerosol in aqueous environments) such as chloroplasts (Gregory and Raps, 1974) and dinoflagellates ( Shapiro et al., 1990, 1991). These particles are also commonly found in the atmosphere and are considered biogenic marine aerosols. For case of dinoflagellates, Liu & Kattawar, (2013) used an ADDA code to simulate the helicoidal shape of the chromosome (a

chiral structure) in the phyoplankton to calculate the 16 Mueller matrix elements. They found a non-zero $S_{14}$ element in backscattering conditions consistent with previously reported observations. They suggested backscattering observations of $S_{14}$ as a possible method to detect dinoflagelates present in red tides. For atmospheric applications, it should be noted that hydrosols are ejected or are precursors of atmospheric biogenic organic aerosols (BOA) in the marine environment, and they exhibit chiral signatures. For example,…

**Lines 360-366, "For example, … overlooked": Please rephrase.**
We agree the paragraph was confusing , we re-wrote it as follows:

OLD:
"For example, organic films containing amino acids attach to ejected marine aerosols and were detected and classified according to their chiral structure (Kuznetsova et al., 2005; Wedyan and Preston, 2008). Further, they do have spectro-polarimetric signatures.
As far as specific reports of observations of CP in atmospheric aerosols or presence of aerosols with chiral elements, the available evidence does indicate that biogenic aerosols contain chiral materials which suggests they should have distinctive CP signatures. Although polarized light has been proposed as a tool to improve visibility observation in the atmospheric marine boundary layer (Quinby-Hunt et al., 1997), the concept of CP detection was seemingly overlooked."

NEW
"For example, Kuznetsova et al., (2005) and Wedyan and Preston, (2008)  measured the chirality of amino acids contained in organic films of ejected marine aerosols.
        With regards to reports of observed CP in atmospheric aerosols, the available evidence is limited because there are very few studies specifically focusing on the V Stokes parameter. For example, although polarized light has been proposed as a tool to improve visibility observations in the atmospheric marine boundary layer (Quinby-Hunt et al., 1997), the concept of using CP for the purposes of detection has been overlooked. However, as we show, there is a reasonable amount of evidence that biogenic aerosols do contain chiral materials, and this suggests these aerosols should have distinctive CP signatures."

**Caption of Fig. 6: Provide more details on what we see in the plots. At which heights the "particle concentrations are rather high" and the "aerosol composition is consistent with organic aerosols"? Discuss the effect of particle hydration below 1-1.5km.**

Caption of Fig 6 does not have such text. We interpret you mean lines 391-394

**Lines 391-394, "Figure 6… (Lee et al., 2018).": The circular polarization is low in the PBL, probably due the high RH and the hydration of the aerosol particles. Moreover, if I**

**understood correctly form the plots, high values of CP are found for low particle concentrations (higher up). Re-write this part, being more specific on what we see in Fig. 6.**

We agree that the description and explanation provided for figure 6 is not adequate and it can lead to misinterpretation. For example, figure 6 shows low Depolarization , not low polarization as the reviewer points out. So, we added much more information and put more context to interpret the image. We also slightly changed our conclusions in relation to the figure. We also edited the figure caption to clarify how to interpret the observed depolarization CP.

OLD

"Perhaps one of the more compelling observations of aerosol's circular polarization in the Earth's atmosphere is reported by Petäjä et al., (2016) . The BAECC (Biogenic Aerosols—Effects on Clouds and Climate) Campaign was carried out during Spring and Summer, 2014 in the Finish arctic forest, and it focused on characterizing the role of biogenic aerosols in cloud formation. The campaign included several examples of collocated surface lidar observations and aircraft overpasses at a time of the year where pollen and other biogenic aerosols are abundant. Figure 6 shows an example from this study where distinctive CP (here defined as ratio of left circular return divided by right circular return) was measured by ground-based HSRL polarization lidar with in-situ confirmation of high aerosol concentrations and composition consistent with organic aerosols in the boundary layer (Lee et al., 2018). "

NEW

"Perhaps one of the more compelling observations of aerosol's circular polarization in the Earth's atmosphere is reported by Petäjä et al., (2016). The BAECC (Biogenic Aerosols—Effects on Clouds and Climate) Campaign was carried out during Spring and Summer, 2014 in the Finish arctic forest, and it focused on characterizing the role of biogenic aerosols in cloud formation. The aircraft deployed included in-situ composition measurements and it operated near a ground based HSRL lidar that specifically measured circular polarization (personal communication, Robert Holtz , lidar operator) as opposed to linear polarization, the more standard way to observe pollen with lidars (Sassen, 2008; Shang et al., 2022). The campaign included several examples of collocated surface lidar observations and aircraft overpasses at a time of the year where pollen and other biogenic aerosols are abundant. The resulting dataset is unique in that it contains remotely observed CP along with in-situ confirmation of biogenic aerosols, known to contain chiral molecules.  Figure 6 shows an example from this study where aircraft in-situ instrumentation data and the corresponding lidar profile are displayed. The left column panels (A and C) show vertical profiles of high CCN and aerosol concentrations in the boundary layer (panel A) and composition measurements demonstrating the high organic carbon content (green line in Panel C) in the boundary layer, consistent with biogenic aerosols (also confirmed by  Lee et al., (2018) in this campaign).  Panel B shows the lidar profile (from surface) of backscattering with an in situ profile of aerosol concentration and both confirm the presence of high concentrations in the boundary layer. Panel D displays a profile of circular

depolarization where a low value implies high circular polarization was observed. In this case, high CP values are reported in the boundary layer where the in-situ data confirms the presences of biogenic aerosol. Petäjä et al., (2016) interprets the high CP as a manifestation of high LP expected if the particles are spherical which may be the case since ambient water content, and thus humidification, was high. Although it is not explicitly mentioned, Petäjä et al., (2016) make use of an expected correspondence between high CP with high LP as expressed by an equation derived by Mishchenko & Hovenier, (1995) (MH95 for short). This is reasonable since there are not many lidar measurements of aerosol CP and the authors use this equation to relate the observed CP with the better known and characterized lidar LP. This expression is frequently used in the lidar community to transform between lidar derived depolarization LP and CP (Robert Holtz, personal communication). While the inference offered by Petäjä et al., (2016) is consistent with the state of knowledge of the time, an alternative interpretation is that CP is the result of the presence of chiral molecules in the biogenic aerosols which, as noted in previous sections, do exhibit CP. "

**Lines 399-400, "While these two… circular polarization.": Please provide more explanations on why the study of Cao et al. (2011) showed no optical activity for bioaerosols (e.g. due to noisy measurements?). These results contradict a big part of the discussion in the manuscript, thus a thorough explanation should be provided here.**

In the backscattering regime, a deviation from the Mischenko and Hovenier (1995) (or MH95) relationship (equation 9 in MH95) would certainly imply a presence of chiral particles or particle alignment.  The latter is what was suggested by Kolokolova and Nagdimunov (2014) , a theoretical study. In our search of experimental proof of the MH95 equation, we only found Cao et al (2011) ( https://doi.org/10.1117/1.3657505) as an example of specifically addressing the validity of the equation. As noted , Cao et al (2011) successfully verified the MH95 equation. We do not know why no optical activity was observed. Certainly, their figures 6, 8 and 9 in Cao et al (2011) are convincing (although at the same time there are not that many points in each plot). One physical explanation could be that aging in the pollen used in their experiment did not contain enough chiral material. As noted in the main text (Salma et el 2010), the chiral nature of biogenic aerosol diminishes through oxidation, and it is conceivable that pollen used in the laboratory setting in Cao et al was just too old (they do not report the freshness of the pollen used). While not specifically related to aerosols, Lucas Patty et al, 2017 found that chirality in live leaves immediately diminishes when is detached from the plant until it becomes negligeable. Another possibility is low sensitivity as the R1 notes, but our combined knowledge is not adequate to provide a critical assessment of the conditions and technical features of the observations reported in Cao et al, (2011).

In addition to CP, collocated and simultaneous linear polarization of measurements would be desirable in a lidar setup (perhaps something like Kokhanenko et al, 2021).  Inspection of the derivation of MH95 equation reveals that randomly oriented chiral particles assumption is needed to use symmetry arguments that cancel elements in the Mueller matrix, and this obtain

such equation. Violation of this rule invalidates the derivation of the equation and suggest the presence of mirror image particles.

It should be noted that the physical setting may be more complicated. Broadly, there are two candidates for chirality. One would be circular dichroism stemming from molecular chirality as already demonstrated by molecular biology and stereochemistry studies cited in the text. The second one may originate from the shape of the pollen and other biogenic particles present . Chiral shapes in pollen are certainly a possibility, but the random orientation of these chiral shapes would probably have result in an ensemble null-$S_{41}$ and $S_{42}$ in the Mueller matrix (unless of course these particles are aligned for some reason). Thus, circular polarization is more likely because of chiral molecules in the aerosol, rather than the aerosol shape itself. Whether this is observable with remote sensors has yet to be determined (other than the case in Fig. 6), and one of the conjectures of this review is that such a signal should be observable.

We changed the respective paragraph and added a bit more discussion on this aspect:

OLD

"Interestingly the distinctive CP found by lidar is in agreement with the theoretical modelling study by Kolokolova & Nagdimunov, (2014) where optically active particles were shown to have non-zero CP and zero linear polarization in the backscattering direction. However, a controlled study (Cao et al., 2011) measuring the degree of LP and CP in pollen backscattering found that both scale with each other following the predictions of Mishchenko & Hovenier, (1995). That study concluded there is no additional aerosol information by measuring both LP and CP. While these two offer somewhat conflicting conclusions, both highlight that indeed biogenic aerosols do produce circular polarization. Clearly additional studies need to be carried out on the independence of information brought by the fourth Stokes term."

NEW

"Interestingly, the distinctive CP found by lidar is in agreement with the theoretical modelling study by Kolokolova & Nagdimunov, (2014) where optically active particles were shown to have non-zero CP and zero linear polarization in the backscattering direction. However, a controlled study (Cao et al., 2011) measuring the degree of LP and CP in pollen backscattering found that both scale with each other following the predictions of Mishchenko & Hovenier, (1995). That study concluded there is no additional aerosol information by measuring both LP and CP. While the authors did not set out to seek CP signals in biogenic aerosols, there could be plausible reasons why no positive CP was observed. These include aging of the pollen (chirality disappears with aerosol senescence as shown by Salma et al, 2010 and in decaying leaves Lucas Patty et al, 2017), not enough representative samples and lack of instrument sensitivity. This result should be confirmed as it appears the only one in the literature found measuring both atmospheric aerosol LP and CP simultaneously. While these two offer somewhat conflicting conclusions, both highlight that indeed biogenic aerosols do

produce circular polarization. Clearly additional studies need to be carried out on the independence of information brought by the fourth Stokes term."

Kokhanenko, G. P., Balin, Y. S., Klemasheva, M. G., Nasonov, S. V., Novoselov, M. M., Penner, I. E., and Samoilova, S. V.: Scanning polarization lidar LOSA-M3: opportunity for research of crystalline particle orientation in the ice clouds, Atmos. Meas. Tech., 13, 1113–1127, https://doi.org/10.5194/amt-13-1113-2020, 2020.

**Lines 405-406, "…there are no studies… the atmosphere": There is the study of Petaja et al. (2016). Please rephrase.**

We rephrased the sentence as follows:

OLD
"However, while these studies stress the fact that atmospheric aerosols do contain OA components, they are aerosol in-situ composition and identification studies and to the extent of this search, there are no studies focusing on the spectral or polarimetric properties and the optical features of bulk chiral aerosols in the atmosphere."

NEW
"However, while these studies stress the fact that atmospheric aerosols do contain optically active components, most of the studies highlighted here were conducted with in-situ instrumentation in the field deployments or in controlled laboratory settings. To the extent of our research, we have found only one study (Petäjä et al., 2016) were remote sensing instrumentation in combination with in-situ (essential to independently confirmed presence of biogenic aerosols) was utilized to detect circular polarization in aerosols. "

**Lines 418-420, "…as well as… matrix.": Replace with "as well as consideration of the effect of chiral or oriented particles on the extinction matrix, resulting in the dichroic extinction of the propagating light."**

DONE

**Lines 441-448, "The size distribution… to be spherical.": Do these aerosol models correspond to specific aerosol types? Please discuss.**

We added more information to make this point clearer.

OLD:
" The simulated case assumes the following conditions: aerosols with very low to high concentrations, homogenously vertical distributed from the surface to 1km, and bi-lognormal size distributions with varying coarse mode concentrations."

NEW

" The general aerosol model setup is for a biogenic marine aerosol with a dark background surface.  The aerosol models chosen for the simulations are generic in the sense they are consistent with marine aerosol models with bi-lognormal size distributions. The variability of composition and size is studied by considering two sets of simulations: one with fixed bilognormal distribution and variable moderate absorption and another with a variable fine mode fraction (FMF) of the total number of particles and fixed refractive index. The goal is to capture a reasonably wide optical range in aerosol size (as expressed by variable FMF) and different refractive indexes. Instead of explicitly considering parameterizations as a function of wind and ambient relative humidity (both known to impact atmospheric radiances through changes in size and index of refraction), we choose a more practical approach by prescribing the FMF and refractive index with a wide range of values expected to be present in marine biogenic aerosols. Parameters selected here were chosen to be consistent and within ranges of the marine aerosol models reported in Sayer et al., (2012). Thus , the simulated case assumes the following conditions…. ".

**Lines 496-497, "However, … by a factor of 2-3.": Provide explanation why**.

In the Rayleigh regime (AOD=0 in the plots), DLP peaks at 90 degrees scattering angle and it decreases towards 180 degrees. This is the case when comparing Rayleigh DLPs between Figures 7 and 8 (note that the labeling of the scattering angle was reversed between in both figures in the submitted doc, now corrected). So, the increase in DLP at a scattering angle closer to 90 can be explained by the molecular contribution to total scattering. Further evidence for this is in the increase with shorter wavelength, where Rayleigh scattering dominates. On the other hand, minimal DCP is present in the Rayleigh only case (AOD = 0), and the factor of 2-3 increase in DCP at the larger scattering angle indicates that it is the nature of the aerosol phase function driving this change. The strong viewing geometry dependence of polarization is expected, so measurements at multiple angles are often made in DLP remote sensing to fully characterize the aerosol state (e.g. Deschamps et al., 1994). Ideal remote sensing with DCP would also presumably make use of multi-angle observations.

Deschamps, P. Y., Bréon, F. M., Leroy, M., Podaire, A., Bricaud, A., Buriez, J. C., and Seze, G.: The POLDER mission: Instrument characteristics and scientific objectives, IEEE T. Geosci. Remote, 32(3), 598—615 , 1994.

**Line 504, "… this is the range of concentrations most globally prevalent…": I do not think this is necessarily true for all aerosol types. Please provide reference to support this statement.**

The global distribution of aerosol concentration or loading as measured by satellite aerosol optical depths is in the 0.05-0.15 range. For example, Levy et al, 2015, figure 8 report such values (https://doi.org/10.5194/amt-8-4083-2015) or figure 1 in Knobelspiesse et al., (2012) (https://doi.org/10.1364/OE.20.021457) or figure 3 in Remer et al, 2006 (https://doi.org/10.1029/2007JD009661) or Anderson et al 2013 with many figures (https://doi.org/10.3402/tellusb.v65i0.20805). There are more modern studies that confirm this and they can be easily verified by looking at the citations of these papers.

We added these references to the respective paragraph to support this point.

**Line 615, "… in Sij is assumed and omitted for simplicity.": Replace with "… in Sij is assumed and omitted for simplicity in Eq. A2 and A3."**

DONE

**Line 632, "… resulting in:": Replace with "… resulting in Eq. A3."**

DONE

**Line 634: Number the equation as Eq. A3.**
DONE

---

## Author Comment (AC2)

**Given that this a manuscript is primarily a review article I approached it as a guide that could be used to justify and drive future developments in the field of polarizations measurements, in general and circular polarization, in particular. Althuogh the majority of my experience is with in situ sensor technology, the measurenments of polarized, scattered light is of particular interest to me for all the reasons that the authors highlight.**

**Reviewer #1 has done a stellar job of identifying a number of areas that need clarification or correction, so there is no point in my reiterating those points, especially since the reviewer caught quite a few that I had missed.**

**I am mostly writing to encourage the authors to address two questions that I think are relevant to this discussion.**

1) **There is much information that can be extracted from both LP and CP measurements, but such information seems to be predicated on knowing a prori if the particles are spherical with homogeneous composition. In the active sensing realm of measurements, measuring the LP components, the polarization ratio is sensitive to the complex refractive index (RI) as well as the asphericity but changes in the ratio due to asphericity tend to be larger than changes due to the CRI.  For measurements of CP in the current application, how can you differentiate changes due to shape compared to changes due to chemistry?**

R2 raises an interesting point for which we do not have a clear answer. Active remote sensing is an advantage because the control in the polarization in the incident light. From the viewpoint of observations (specifically of real atmospheric aerosols), we have not encountered much evidence regarding CP. For LP , it is clear that shape and size drives the magnitude of DLP more than composition. For CP, there is not enough evidence to say one way or another. It is possible that in the realm of polarized optical cytology research (detection of biological cells) particularly in the sub-field of stereochemistry, this question has been addressed. We found a large body of knowledge in sophisticated optical modelling, observation and attempts to distinguish non-spherical cells (many are cited in section 3). However, it is difficult to tell to what degree lessons learned in this research area are directly applicable  to atmospheric applications, at first sight the mediums (blood plasma in one discipline contrasted with air) and composition (cells and atmospheric aerosols) are just too different.

In section 5, we had the following paragraph :

"Interestingly the distinctive CP found by lidar is in agreement with the theoretical modelling study by Kolokolova & Nagdimunov, (2014) where optically active particles were shown to have non-zero CP and zero linear polarization in the backscattering direction. However, a controlled study (Cao et al., 2011) measuring the degree of LP and CP in pollen backscattering found that both scale with each other following the predictions of  Mishchenko & Hovenier, (1995). That

study concluded there is no additional aerosol information by measuring both LP and CP. While these two offer somewhat conflicting conclusions, both highlight that indeed biogenic aerosols do produce circular polarization. Clearly additional studies need to be carried out on the independence of information brought by the fourth Stokes term"

where we attempt to stress that within the little evidence we found, there is seemingly contradictory evidence. But since these studies are so different and there are so few papers focusing on CP, we just can't conclude one way or another. We chose not to add more discussion as it is a bit speculative without additional observational evidence.

An additional point is that passive observations of LP are often made at multiple viewing angles, which provide for sampling of different portions of the aerosol phase function and thus the means to differentiate in a way that cannot be done at backscattered directions alone. While presumably such an approach would be valid for CP as it is for LP, the literature on multiple angle CP measurements is scant.

In summary, we think this is an open question as far as CP and more dedicated studies are needed. We are hopeful that with all the technical advances in the last few years observations of the Full Stokes vector will be become more mainstream and thus would inspire more explorations on the information content in CP.

2) **Athough this review paper specifically discusses CP produced from interactions of unpolarized light with particle ensembles, I think it might have been useful to take some examples from the lidar community who already employ CP to show how these measurements are already highlighting how CP measurements offer complementary information to LP.**

It appears the reviewer is aware of CP observations that we were not aware or have not access to. In the gathering of information for this review, we found that there are just a handful of studies dedicated to measure circular polarization in Earth sciences applications, particularly aerosols. In fact, we found just one observational study (Petjaa et al, 2014, cited) where CP scattered by aerosols was observed and, in this case, an HSRL lidar with an incident polarized laser and CP received was used. This study was singular in that it had independent confirmation of presence of biogenic aerosols. In contrast , there are numerous studies focusing on the measurement aerosol linear depolarization with lidar. In conversations with lidar investigators, we found several reasons why circular polarization observations have not been reported. One of them is that the theoretical study by Mishchenko and Hovenier (1995) (cited) demonstrated that in the backscattering applications (ie. lidar), DLP and DCP are not independent, and one can be derived from the other. We downloaded the lidar data from the Petjaa et al paper and files included both CP and LP backscattering measurements. This seemed contradictory to the

statement in the same paper that only CP was measured. We approached the lidar PI to ask the source of LP data in the lidar files. He confirmed that indeed CP backscattering was measured, and they used the MH equation to derive DLP. He said that because DLP is a more commonly used parameter in the lidar community, it made sense to make it available. This illustrates the attitude in the lidar community of the current usage of polarized observations and why LP seems more favored. Further, because CP is (normally) at least an order of magnitude lower than LP, it has been a technical and operational challenge to attempt CP measurements in the field for many groups. (A version of this explanation has been added prompted by R1 comments. See new text in Section 5 where the data in figure 6 is described)

As result, there have not been many studies focusing on CP associated to atmospheric aerosols at all. From these conversations, it became clear to us that these reasons were part of the general wisdom by which many lidar researchers operate. There has been a few more explorations regarding CP measurements in clouds but we did not dwell much in it. Thus, it is hoped that this work provides an incentive to explore additional polarization properties in atmospheric aerosols.

So, in addressing the R2 question, there are just not enough observations of atmospheric CP (to the extent of our literature research) that can provide a clear an indication one way or another. If the reviewer is aware of such information, we would very much appreciate if he/she could share it so we can acknowledge it.

**I think that the authors have done a very comprehensive job of promoting the benefits of CP measurements and this will be a very useful review that will be used for developing future measurement capabilities.**

We sincerely  thank the reviewer for the kind words.

---

## Author Comment (AC3)

**Summary:**
The paper presents an literature review about circular polarization. It explains how circular polarization is generated and which aerosols in the atmosphere produce circular polarization. A few radiative transfer simulations are carried out to investigate the sensitivity of circular polarization on aerosol optical thickness and on aerosol composition. The conclusion is that measurements of circular polarization from space could provide additional information on aerosol compared to measurements of intensity and linear polarization only. The literature review is very interesting to read but it does not include new scientific results. The simulation part is very short and also seems to be not correct (see comment below). For these reasons I can not recommend to publish the paper in its current status, in my opinion major revisions are required.
**General:**
- It is claimed that technical advances have demonstrated the feasibility to measure the full Stokes vector from space. However, this is not discussed in the text in more detail. The literature review and also the simulations show, that circular polarization (CP) is several orders of magnitude smaller than linear (LP). The authors should provide more information about planned passive instruments that could measure CP with such a high precision from space.
- The RT simulations are not convincing, because they do not include absorption by trace gases (e.g. ozone in UV). The results show higher linear polarization for a scattering angle of 160.8° than for a scattering angle of 112°, which is not expected because the maximum polarization for Rayleigh scattering is at 90° scattering angle. The relative sensitivity of CP to aerosol composition is large, but since the magnitude is so small it will still be very difficult to measure it.

Thank you for your review. The primary purpose of this manuscript is to review the state of the literature pertaining to CP. Such reviews are within the scope of the journal ACP, and there are many recent examples (e.g. Gao et al., 2021, Bright and Lund, 2020, Korolev and Leisner, 2020). The simulations described in the manuscript were included to augment gaps in the literature and further strengthen the case for additional investigations of CP. In any case, the need for 'new scientific results' is not an appropriate metric for the acceptance or rejection of a manuscript in ACP when it is intended to be a review. That said, you did help us notice a mistake in our simulations that we have since corrected. This is described in more detail below.

Answer to General Comments:

**Point 1:** We respectfully  disagree. As a review that introduces a new subject to the wide aerosol community (as it is the case for ACP), we dedicated an appropriate amount of discussion to measurement feasibility. To some extent, this

review is a call for additional measurements as well. Certainly there are more aspects to explore and we could have dedicated additional space to discuss with more detail the detection capabilities of a number of instruments that are currently being planned or in design stages. But it would have lengthened the manuscript in a way that would have distracted the readership from the main point of the paper, which is the inherent potential of CP to reveal the physical nature of aerosols.

**Point 2:** While we agree that RT computations could have been more thorough and realistic (such as including gas absorption), it would have been beyond the scope of a manuscript, and are only meant to illustrate the general features of circularly polarized radiances in aerosols. This type of study has never been published before. Perhaps the closest paper to this study is the Slonaker et al (2005, cited), a conference paper. In this paper and to make a similar point, they did not use trace gas corrections nor considered sensor bands despite that they do discuss simulations for satellite observations. Most sensor systems operating in the UV / Visible / Near Infrared choose spectral bands to be in windows with minimal or correctable trace gas absorption. Our simulations are meant to replicate this characteristic shared with successful aerosol remote sensing missions such as MODIS, VIIRS and POLDER. The simulation figures are meant to express the changes in intensity, LP and CP as a function of matters such as geometry and aerosol load, changes which are clear even without accounting for minimal trace gas absorption. Again, we emphasize that a more dedicated study would be desirable, but this should be the subject of a specific study and published separately. Regarding the issue of angles, we found an error in the equation used to compute the scattering angle. When corrected, the expressed linear polarization is higher closer to a scattering angle of 90˚ in line with expectations for Rayleigh scattering. Thank you for helping us realize this error.

Slonaker, R. L., Takano, Y., Liou, K.-N., & Ou, S.-C. (2005). Circular polarization signal for aerosols and clouds. In H.-L. A. Huang, H. J. Bloom, X. Xu, & G. J. Dittberner (Eds.), *Proc. SPIE* (Vol. 5890, pp. 58900B-58900B–8). https://doi.org/10.1117/12.619576

**Specific comments:**
**l. 165 ff: "For example, exact solutions can be found for 2pr/l << 1 and |mr/l|<< 1 (Rayleigh scattering) ..."-> you are talking here about the scattering matrix elements S_4j and S_i4, which describe circular polarization. For Rayleigh scattering they are exactly 0, because Rayleigh scattering does not cause any circular polarization.**

Yes, it is a bit confusing, these are two sentences describing different points. We added a small correction to separate the concepts expressed.

OLD:

"….. For example, with incident unpolarized light (represented as $I_{in}$[1,0,0,0]) in Equation 1, the system will exhibit CP if the matrix has non-zero $S_{4j}$ and $S_{i4}$ that result in an the outgoing vector with non-zero $V_{out}$. From the modelling viewpoint, there are very few analytical solutions of the Maxwell's equations that provide the sixteen $S_{ij}$ coefficients. For example…. "

NEW:

"….For example, with incident unpolarized light (represented as $I_{in}$[1,0,0,0] in Equation 1) onto aerosols, the system will exhibit CP if the matrix has non-zero $S_{4j}$ and $S_{i4}$ that result in an the outgoing vector with non-zero $V_{out}$.
       From the modelling viewpoint, there are very few analytical solutions of the Maxwell's equations that provide the sixteen $S_{ij}$ coefficients. For example…. "

**Fig. 3 (caption): "right panel" should be "left panel" and reference to "right panel" is missing**
DONE

**l. 298: "This concept has been applied in astronomy studies where the observations of linear and circular polarization in comets and interstellar dust" -> What is the observed degree of linear/circular polarization in interstellar dust?**

The Rosenbush et al, 2007 reported CP up to 0.8% in comets. In the introduction of Whitnet and Wolff (2002, cited) , several observational papers are cited  with observed CP ranging 1-17% in dust associated to protostars.

**l. 325: "Lidars with CP detection capabilities have been proposed for cloud phase" -> How can cloud phase be detected with circular polarization? Please explain.**

The cited papers in this sentence show modeling studies of linear and circular polarization lidar observing scenarios for incident polarized signals. They show that the different shapes in ice and liquid particles result in both LP and CP.  This is explained in the cited references, and we do not think it is relevant to put this additional information here as this is an aerosol focused review.

**l. 329: "... (Gilbert and Pernicka, 1967; Lewis et al., 1999) and foggy atmospheres (van der Laan et al., 2017). These studies highlighted the fact that in high- and low-density particle environments, the propagation of circular polarization does not degrade as quickly as linear polarization." -> Why should linear polarization degrade quickly in low-density particle environments? Why does circular polarization degrade slower than linear?**

We agree with the reviewer that this is a bit misleading. The papers cited are concerned with very turbid environments and the simulations presented in those studies are for high optical depths (>1) , large particles (fog and marine sediments) and for incident light with a defined polarity (ie incident LP or CP).  One paper (Gilbert and Pernicka, 1967, cited) reports laboratory observations.  The van der Laan et al (2018) study (not cited) make this point very clear (see respective figures 4 and 12) . They show that the persistence of DCP with respect to  DLP as a function of AOD in visible and NIR wavelength ranges. The statement regarding loss of LP in low density environment is in the context of very turbid environments which is not the case of interest here. Also, note the van der Laan et al (2018) computations do not include the range AOD =0 to 1. This is the range considered in our study. The RT simulations shown in section 6 are the first time to our knowledge where this range of aerosol concentrations are considered.

We modified the respective paragraph to make this clearer:

OLD
"…… and foggy atmospheres (van der Laan et al., 2017). These studies highlighted the fact that in high- and low-density particle environments, the propagation of circular polarization does not degrade as quickly as linear polarization. "

NEW
"…… and foggy atmospheres (van der Laan et al., 2018). These studies highlighted the fact that in high-concentration environments (AOD>1), the propagation of circular polarization does not degrade as quickly as linear polarization."

J. D. van der Laan, J. B. Wright, S. A. Kemme, and D. A. Scrymgeour, "Superior signal persistence of circularly polarized light in polydisperse, real-world fog environments," Appl. Opt., vol. 57, no. 19, p. 5464, Jul. 2018.

**l. 395: "inagreement" -> "in agreement"**
DONE

**l. 395: "Interestingly the distinctive CP found by lidar is inagreement with the theoretical modelling study by Kolokolova & Nagdimunov, (2014) where optically active particles were shown to have non-zero CP and zero linear polarization in the backscattering direction. However, a controlled study (Cao et al., 2011) measuring the degree of LP and CP in pollen backscattering found that both scale with each other following the predictions of Mishchenko & Hovenier, (1995). That study concluded there is no additional aerosol information by measuring both LP**

**and CP. While these two offer somewhat conflicting conclusions, both highlight that indeed biogenic aerosols do produce circular polarization" -> What are the different assumptions in the studies by Kolokolova and Nagdimunov 2014 and Mishchenko&Hovenier 1995? I assume these are theoretical studies, so there should be a simple explanation for the conflicting conclusions?**

There are contrasting differences between both studies, and it is not entirely clear the attribution of the source of those differences. But just to mention a few, the MH95 paper uses simple spherical particles, and it takes advantage of the symmetry in the scattering matrix at the backscattering angle. The KN14 study uses aggregates of small spheres made of chiral material and it does not use symmetry arguments in the calculations. They used a rather sophisticated code (a multi-sphere T-Matrix code, from Mackowski et al., 2011) to calculate the optical properties to report results whereas MH95 only reports calculations from simple equations. We do not think this is the place to elucidate why as we can only speculate without doing a more in-depth study.

However, R1 raised a similar comment regarding this section and the lack of additional explanations. We expanded this section additional text. We refer to the respective answer to R1 for more context. Here we add the new paragraph in this section:

OLD

"Interestingly the distinctive CP found by lidar is in agreement with the theoretical modelling study by Kolokolova & Nagdimunov, (2014) where optically active particles were shown to have non-zero CP and zero linear polarization in the backscattering direction. However, a controlled study (Cao et al., 2011) measuring the degree of LP and CP in pollen backscattering found that both scale with each other following the predictions of Mishchenko & Hovenier, (1995). That study concluded there is no additional aerosol information by measuring both LP and CP. While these two offer somewhat conflicting conclusions, both highlight that indeed biogenic aerosols do produce circular polarization. Clearly additional studies need to be carried out on the independence of information brought by the fourth Stokes term. "

NEW
"Interestingly the distinctive CP found by lidar is in agreement with the theoretical modelling study by Kolokolova & Nagdimunov, (2014) where optically active particles were shown to have non-zero CP and zero linear polarization in the backscattering direction. However, a controlled study (Cao et al., 2011) measuring the degree of LP

and CP in pollen backscattering found that both scale with each other following the predictions of  Mishchenko & Hovenier, (1995). That study concluded there is no additional aerosol information by measuring both LP and CP. While the authors did not set out to seek for CP signals in biogenic aerosols, there could be plausible reasons why no positive CP was observed. These include aging of the pollen (chirality disappears with aerosol senescence as shown by Salma et al, 2010 and in decaying leaves Lucas Patty et al, 2017), not enough representative samples and instrument sensitivity.  This result should be confirmed as it appears (to the extent of our search) the only study found in the literature measuring atmospheric aerosol LP and CP simultaneously. While these two studies (one theoretical, the other experimental) offer somewhat conflicting conclusions, both highlight that indeed biogenic aerosols do produce circular polarization. Clearly additional studies need to be carried out on the independence of information brought by the fourth Stokes term."

Mackowski, D., Kolokolova, L., & Sparks, W. B. (2011). T-matrix approach to calculating circular polarization of aggregates made of optically active materials. Journal of Quantitative Spectroscopy and Radiative Transfer, 112(11), 1726–1732. https://doi.org/10.1016/j.jqsrt.2011.02.003

**l. 429: Eq. 4 (circular polarization after 2 scattering events) -> This equation is not very special (included in all VRT codes handling circular polarization). Why is this equation derived and written down here, it is not used at all in the discussion of the results?**

We do think that this equation is special in the sense that it provides one of the few analytical ways to illustrate with an equation how CP is linked to incoming radiation. The fact that is included in RT codes does not make it any more accessible or understandable to the scientific community. We included this equation because for much of the intended audience of this paper may not be as versed in the intricacies of RT codes and for whom, polarized radiative transfer computations are generated by complicated modeling tools.

**l. 447: "The particle shape is assumed to be spherical." -> In the discussion you highlighted the importance of particle shape and orientation for circular polarization. Then, in the model simulations it is neglected. At least randomly oriented aspherical particles can be handled in most state-of-the-art VRTE models...**

We certainly agree that this would be the ideal case, but this option was not available to us (computer power, reliable non-spherical code) at the time of carrying out the simulations.

**l. 449: "The atmospheric column only contains air (i.e., no trace gases)" -> Why modelled without trace gases? Calculations without O3-absorptions are very unrealistic in the UV-range. To my understanding "air" normally includes also the trace gases.**

We think that as a first attempt to study aerosol CP propagation in the atmosphere at representative wavelengths, this exercise is adequate. Yes, in real observational settings there are absorbing gases and other sources of noise. But as at first of its kind, we wanted to focus on the possible signals. Certainly, additional more realistic scenarios should be tried. We hope these computations are enough to trigger the curiosity to explore this subject in more detail.

**Figs. 7/8: What is the definition of the scattering angle? Normally it is given by cos(theta_s)= cos(n_inc * n_sca), where n_inc and n_sca are incoming and scattered directions, respoectively? I doubt that Fig. 7 is for scattering angle of 160.8 degrees. LP by Rayleigh scattering has a maximum around 90° scattering angle, so I would expect much higher LP in Fig. 8 (scattering angle 112°) than in Fig. 7 (160.8°, backscattering direction).**

We thank the reviewer for bringing up this point. Verifying computations that have never been reported before can be a challenge. As the reviewer points out DLP at near the 90-degree scattering angle should be the highest compared to other scattering angles. This is expressed by equation

$$DLP = (Iper - Ipar)/(Iper + Ipar)$$
$$= sin(theta)^2 / (1+ cos(theta)*cos(theta))$$

where Iper stands for radiance measured perpendicular to the scattering plant and Ipara is the parallel component of the radiance. Theta is the scattering angle as defined in Appendix Figure 1. (See Coulson book, Page 179 , Eq 4.25)

So, we rechecked our computations and we found a mistake in the equation we used to compute the scattering angle (a minus signed was misplaced). After correcting the equation and recomputing the angles, the new angles changed to:

SCAT_ANG(old) = 160.8 changed to SCAT_ANG(new) = 103.2
SCAT_ANG(old) = 112.3 changed to SCAT_ANG(new) = 142.5

These new angles are in accordance to what above equation with DLP(103.2) > DLP(142.5) as figures 7 and 8 show at AOD=0 and demonstrate that our computations are in accordance to what is expected.

Coulson, K. L. (1988). Polarization and intensity of light in the atmosphere. Hampton, Va., USA: A. Deepak Pub.

**l. 522 "Overall, these plots suggest there is a sensitivity to differences between coarse and fine mode dominated aerosols in both linear and circular polarization." -> Since CP is scaled with 10â´, I assume that even if the relative sensitivity is higher in CP, it can still be better measured in LP? I can not see the benifit of measuring CP.**

We disagree with R3 in that we think there is a benefit to explore the measurement of aerosol CP.  Again, while we acknowledge that the RT simulations are not as complete as R3 wished, the evidence shown in the first two thirds of the paper plus these limited (but entirely new) RT simulations suggest when looking at all this information together, there are lines of research of  interest and unexplored in the subject of aerosol polarization. As far as the magnitude of DCP, it is certainly a low magnitude but not null and it is the thesis of this review that technological advances are improving at a rapid pace such that no only CP measurements can be made in laboratory settings but also remote (as shown in the Petjaa et al study cited in the paper). So it is appropriate to start to explore the benefits of measuring CP as it appears suitable technologies are already available .

**Fig.9, labels in right panels:  U->V**
Corrected

**What is the scattering angle of the simulations shown here?**
Same as in figure 8. It is indicated in the respective caption.

**Author contribution: What is the contribution of KK?**
The text corresponding to KK contribution was missing.
OLD
SG carried literature gathering and review, performed the simulations, carried out the analysis, created the figures and wrote the paper.
NEW
SG carried literature gathering and review, performed the simulations, carried out the analysis, created the figures and wrote the paper. KDK reviewed several versions of the manuscript, provided several key suggestions and background material.

---

## Author Response (AR1)

Aug/18/2022

Dear Editor

Please find attached the response to the reviewers for the manuscript entitled "Circular Polarization in Atmospheric Aerosols" ( acp-2022-55) by myself and Kirk Knobelspiesse.

All questions, comments and corrections have been addressed and modifications added to the new manuscript. This version of the manuscript has changed (for the better we think) but not in a major way and the main points and conclusion remain the same.

We would like to thank the three reviewers as all had interesting comments (even the reviewer who recommended major changes and spotted an error in the computations).

Also, we would like to apologize for the delay in submitting the response. Many points required some bibliography research and careful writing, this resulted in a total of 28 pages to answer the reviewers (to compare the actual manuscript is 22 pages without references). Also, one of went on vacation and was a lead organizer of an international meeting, so everything got delayed. We thank the editorial office for being flexible with the deadline.

As I am submitting this answer, I am going on vacation in two weeks so, I will be able to address any inquiry on my return in the first week of September.

Sincerely

Santiago Gassó

Santiago Gassó, Ph.D., GESTAR/NASA
sgasso@umd.edu